# Efficient urea electrosynthesis from carbon dioxide and nitrate via alternating Cu–W bimetallic C–N coupling sites

Yilong Zhao[1,4], Yunxuan Ding[2,4], Wenlong Li[1,2], Chang Liu[1], Yingzheng Li[1], Ziqi Zhao[1], Yu Shan[1], Fei Li [1], Licheng Sun [1,2,3] ✉ & Fusheng Li [1] ✉

Electrocatalytic urea synthesis is an emerging alternative technology to the traditional energy-intensive industrial urea synthesis protocol. Novel strategies are urgently needed to promote the electrocatalytic C–N coupling process and inhibit the side reactions. Here, we report a $CuWO_4$ catalyst with native bimetallic sites that achieves a high urea production rate ($98.5 \pm 3.2\ \mu g\ h^{-1}\ mg^{-1}_{cat}$) for the co-reduction of $CO_2$ and $NO_3^-$ with a high Faradaic efficiency ($70.1 \pm 2.4\%$) at $-0.2\ V$ versus the reversible hydrogen electrode. Mechanistic studies demonstrated that the combination of stable intermediates of $*NO_2$ and $*CO$ increases the probability of C–N coupling and reduces the potential barrier, resulting in high Faradaic efficiency and low overpotential. This study provides a new perspective on achieving efficient urea electrosynthesis by stabilizing the key reaction intermediates, which may guide the design of other electrochemical systems for high-value C–N bond-containing chemicals.

Urea is used in nitrogen-based fertilizers and has supported a large proportion of the crop yield increase and assured the food supply of humanity[1,2]. In industry, urea is synthesized using the Bosch-Meiser process, where $CO_2$ and liquid ammonia are mixed in urea through an ammonium carbamate intermediate under severe high-pressure conditions (150–250 bar) and elevated temperature (150–200 °C)[3]. Moreover, industrial ammonia manufacture (Haber-Bosch method) is highly energy-intensive, consumes considerable amounts of fossil fuels, and heavily emits $CO_2$; thus, the current urea synthetic protocol is far from meeting the demands of society for sustainable development[4]. By contrast, electrosynthesis can convert feedstocks into high-value-added chemicals using renewable energy, being a more sustainable process and enabling the decarbonization of urea production[2,5].

To date, direct activation of $N_2$ coupled to $CO_2$ for urea in ambient conditions is still challenging, because of the high overpotential requirements for the dissociation of highly stable C=O (806 kJ $mol^{-1}$)

and N≡N (941 kJ $mol^{-1}$) bonds, as well as the strong competition of the parallel reactions[6–8]. Reactive nitrogen-oxygen bond-containing species, such as nitric oxide (NO) and nitrate/nitrite ($NO_3^-/NO_2^-$) ions, are more active nitrogen feedstocks. Among these species, $NO_3^-$ is a nitrogen-containing reactant with a better intrinsic instability, which can be obtained from industrial wastewater; or by potentially sustainable nitrate generation technology in the future, such as non-thermal plasma activation of nitrogen[9,10]. Moreover, the lower dissociation energy of the nitrogen-oxygen bond (204 kJ $mol^{-1}$) eases the coupling of $NO_3^-$ reduction with $CO_2$ reduction to accomplish urea electrosynthesis[11]. Thus, the $NO_3^-$ to urea process is a suitable model reaction to study the electrochemical C–N bond formation; and a potential synthetic protocol for urea synthesis.

Since 1998, $NO_3^-$ has been reported as a nitrogen-containing feedstock that can be coupled with $CO_2$ for urea electrosynthesis[12]. However, the complex 16-electron reduction process has restricted

[1]State Key Laboratory of Fine Chemicals, Institute of Artificial Photosynthesis, DUT-KTH Joint Education and Research Centre on Molecular Devices, Dalian University of Technology, 116024 Dalian, China. [2]Center of Artificial Photosynthesis for Solar Fuels and Department of Chemistry, School of Science, Westlake University, 310024 Hangzhou, China. [3]Department of Chemistry, School of Engineering Sciences in Chemistry, Biotechnology and Health, KTH Royal Institute of Technology, 10044 Stockholm, Sweden. [4]These authors contributed equally: Yilong Zhao, Yunxuan Ding. ✉e-mail: sunlicheng@westlake.edu.cn; fusheng@dlut.edu.cn

studies on the development of $NO_3^-$-to-urea[13]; in particular, the comprehensive mechanistic understanding of electrocatalytic C–N bond formation and the structure design of catalysts remain fundamental challenges[14,15]. To date, urea electrosynthesis with $NO_3^-$ and $CO_2$ as feedstocks has reached an FE of up to 53% with a current density of $0.3\,mA\,cm^{-2}$[16]. Furthermore, the applied potentials for efficient urea electrosynthesis range from −0.6 to −1.5 V vs. reversible hydrogen electrode (RHE)[15,16], which are far away from the thermodynamic potential for urea synthesis from $CO_2$ and $NO_3^-$ (0.48 V vs. RHE)[17]. Under such negative applied potentials, kinetically favorable competing side reactions, such as $H_2$ generation, $CO_2$ reduction, and $NO_2^-$, $NH_3$ production, easily occur and reduce the selectivity of urea electrosynthesis[18].

As the primary intermediate of $NO_3^-$ reduction, the formation of $^*NO_2$ does not involve complex elementary reactions that may dissociate various by-products. Thus, $^*NO_2$ serves as a vital N-intermediate that reacts with intermediates of $CO_2$ reduction, which may reduce the overpotential of urea electrosynthesis and reduce the probability of by-product generation. Cyanobacteria can utilize $NO_3^-$ to photosynthetically synthesize organic nitrogen compounds with the help of nitrate and nitrite reductase metalloenzymes. In the first step of nitrate assimilation, nitrate reductase, a Mo-bis-molybdopterin guanine dinucleotide with a high-valence $Mo^{4+}$-based reaction center, can convert $NO_3^-$ into $^*NO_2$ intermediate and rapidly dissociate to $NO_2^-$ at a low reduction potential[19,20]. A relatively long lifetime of N-related intermediates is expected to provide more opportunities for coupling with the intermediates of $CO_2$ reduction, which the C–N bond formation can achieve urea electrosynthesis. The reaction of a high-valence metal center can decrease the electron density of the adsorbed species, resulting in a more positive reaction overpotential[21,22]. As tungsten is a homolog of molybdenum, high-valence tungsten-oxide derivatives are conducive to stabilizing $^*NO_2$ intermediate(s); such as $WO_3$ could strongly adsorb $NO_2$ molecules on the surface[23,24]. However, $WO_3$ itself cannot trigger the $CO_2$ reduction at low overpotentials, because $^*CO$ intermediate is challenging to be formed[25]. Contrarily, $^*CO$ is the common $CO_2$ reduction intermediate on Cu-based catalysts, but continuous Cu sites enable the C–C coupling between adsorbed $^*CO$ and/or $^*CHO/^*COH$ intermediates[26]. For urea electrosynthesis, a higher C–N coupling proportion is expected; separating the Cu sites could reduce the probability of C–C coupling by-products[27].

Based on the considerations above, in this study, a Cu–W bimetallic oxide ($CuWO_4$) catalyst with alternating bimetallic reaction sites was utilized for urea electrosynthesis with $CO_2$ and $NO_3^-$ as feedstocks (Fig. 1). A milliampere-level current of urea electrosynthesis could be realized at a remarkably operating potential with the highest FE reported to date. The reaction pathways and intermediates were systematically studied by in situ Raman spectroscopy and differential electrochemical mass spectrometry (DEMS), demonstrating that the rate-determining step of the urea generation from $CO_2$ and $NO_3^-$ is the $^*NO_2$ and $^*CO$ intermediates hydrogenation and coupling on $CuWO_4$. Combined with thermodynamic adsorption energy analysis and theoretical calculation, the alternating bimetallic sites effectively improve the formation and coverage of the two intermediates on the surface of the catalyst, increasing the probability of C–N coupling and the selectivity for urea electrosynthesis.

## Results

### Catalyst synthesis and characterization

The $CuWO_4$ catalyst was prepared by a hydrothermal synthesis approach using tungstate and Cu salts as raw materials[28]. The X-ray diffraction (XRD) pattern, Raman and Fourier-transform infrared spectrum of the synthesized sample could be assigned to the $CuWO_4$ triclinic structure (Fig. 2a, Supplementary Fig. 1)[29,30]. The X-ray photoelectron spectrum (XPS) of Cu 2p (Fig. 2b) and W 4f (Fig. 2c) exhibited two distinct split spin-orbit peaks located at 934.3 ($Cu^{2+}$ $2p_{3/2}$), 954.2 ($Cu^{2+}$ $2p_{1/2}$), 35.2 ($W^{6+}$ $4f_{7/2}$), and 37.4 eV ($W^{6+}$ $4f_{5/2}$), which indicated that Cu and W were in the II and VI oxidation states[31], respectively. The lattice oxygen (O–Cu or W, 530.1 eV) was observed in O 1s XPS (Supplementary Fig. 2)[32]. The morphology of the $CuWO_4$ catalyst manifested as nanoparticles with a size of 40–60 nm by scanning electron microscopy (SEM, Fig. 2d) and cryogenic

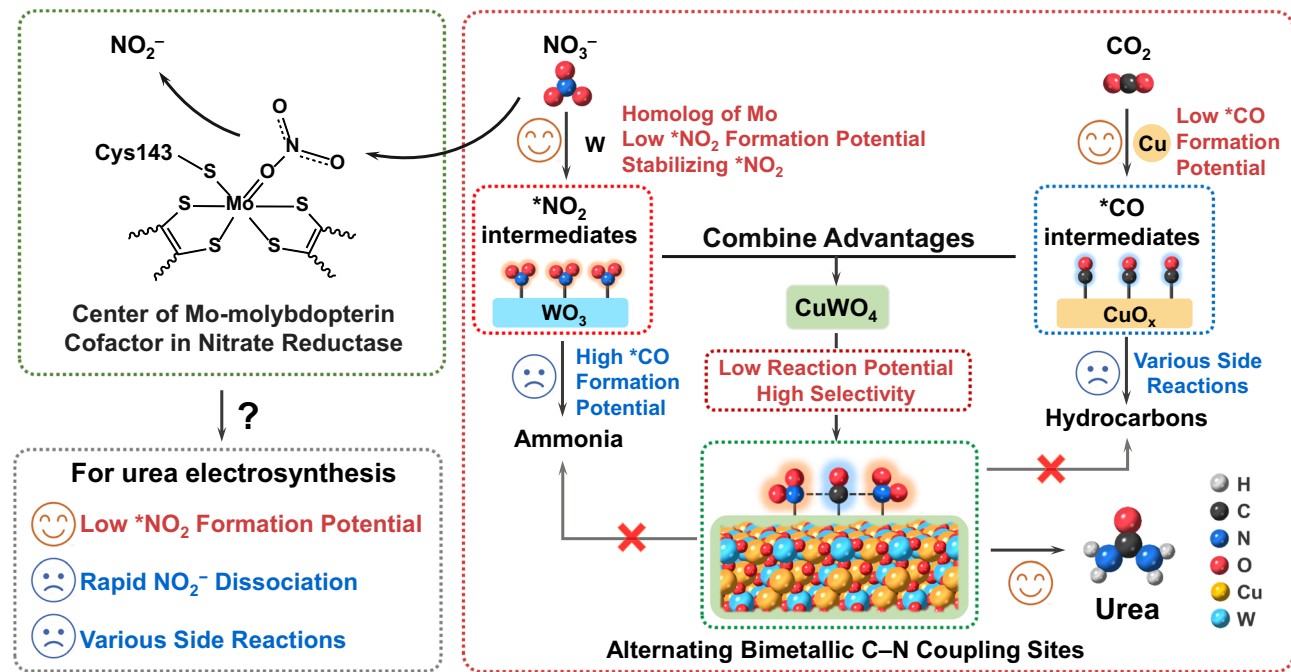

**Fig. 1 | Schematic illustration of design strategy for bioinspired alternating bimetallic sites of $CuWO_4$ catalyst for urea electrosynthesis.** The stabilization of the activated $^*CO$ and $^*NO_2$ intermediates at Cu and W bimetallic sites may play an important role in the high efficiency of C–N coupling.

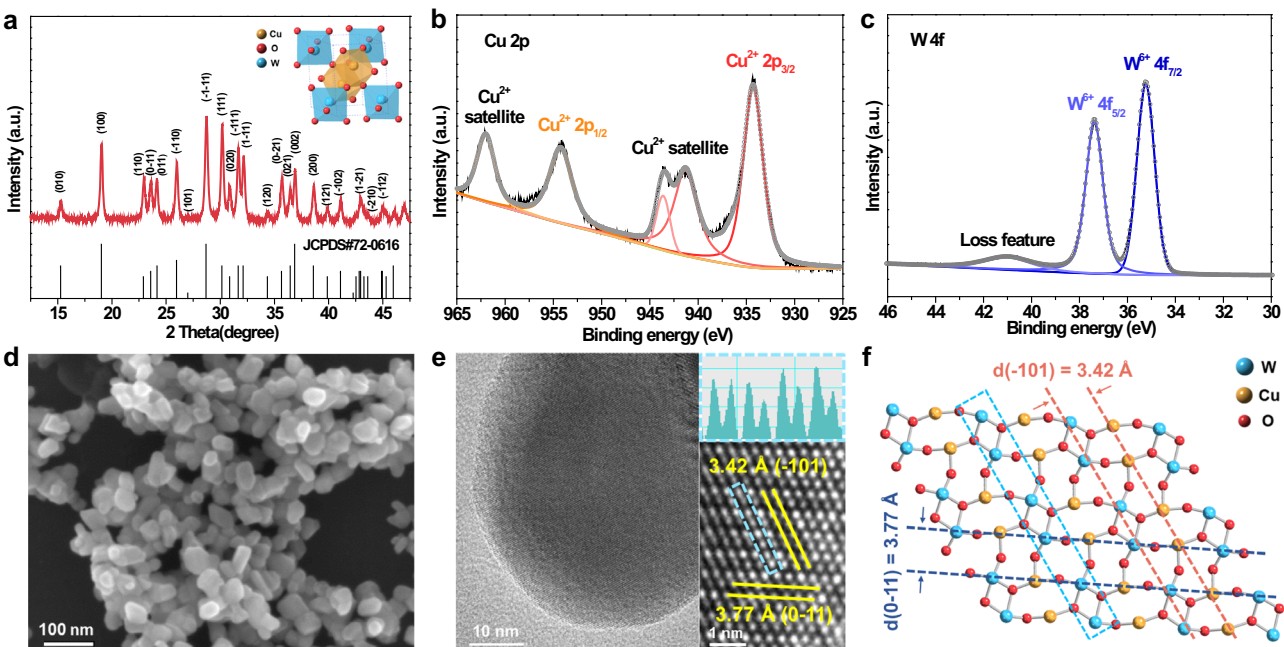

**Fig. 2 | Structural and compositional characterization of CuWO₄ catalyst. a** XRD pattern. Inset: crystal structure of CuWO₄. **b, c** XPS spectra of Cu 2p (**b**) and W 4 f (**c**). **d** SEM image. **e** TEM and atomic-resolution HAADF-STEM image with line profile of the corresponding area. **f** Atomic structure of CuWO₄ (111) facet.

transmission electron microscopy (cryo-TEM, Fig. 2e). Cu, W, and O were homogeneously dispersed throughout the nanoparticle-like catalyst (Supplementary Fig. 3). The entire lattice exhibited highly ordered rectangular arrays with alternating bright and dark columns of atoms (Fig. 2e right), which was revealed by the high-angle, annular dark-field scanning transmission electron microscopy (HAADF-STEM). The transverse and longitudinal array spacings were 3.77 and 3.42 Å, corresponding to the interplanar spacing of (0−11) and (−101) facets, respectively[33]. The spacing between metal atoms in the same longitudinal array showed a weak periodic change. The triclinic CuWO₄ crystal structure was constructed (Supplementary Fig. 4) and exhibited the dominant (111) facet (Fig. 2f), which is consistent with the atomic structure in the HAADF-STEM image. Therefore, the synthesized nanoparticles were highly ordered triclinic CuWO₄ and did not contain other detectable impurities such as other derivatives of Cu or W. Pure CuO and WO₃ catalysts were also prepared with similar morphology and valence state to CuWO₄ for comparison; and were fully investigated by various characterization methods (Supplementary Figs. 5–9).

## Electrochemical urea synthesis

Electrochemical measurements of urea synthesis were conducted to investigate the catalytic performance of CuWO₄ and other samples (Methods, Supplementary Fig. 10). Figure 3a shows the linear sweep voltammetry (LSV) curves and *I-V* plots of CuWO₄, CuO, and WO₃. When switching the atmosphere of the electrolyzer from Ar to CO₂, the reduction current of CuWO₄ and CuO increased at the potential range from −0.1 to −0.4 V vs. RHE, indicating that additional CO₂ reduction-related reactions may occur. For WO₃, no significant difference between the two LSV curves was observed until to the more negative potential region (−0.7 to −0.9 V vs. RHE). For accurately measuring the urea synthesis efficiency of the catalysts at the corresponding potential, the chronoamperometry (CA) method was employed (Methods, Supplementary Fig. 11); and the production of urea was quantified by diacetylmonoxime-thiosemicarbazide (DAMO-TSC) (Supplementary Figs. 12–13) and nuclear magnetic resonance (NMR) methods. As the UV-visible spectral of DAMO-TSC method could be affected by the variation in the concentration of nitrite (Supplementary

Figs. 14–17)[34,35]. Therefore, the urease decomposition method was further adopted and calibrated by two ammonium ion quantification methods (Supplementary Figs. 18–19). Other N-based by-products such as nitrite, ammonia, and hydrazine were quantitatively analyzed using ion chromatography, indophenol blue, and Watt and Christo's methods, respectively (Methods)[35,36]. Gas-phase by-products were analyzed by gas chromatography (GC) (Supplementary Figs. 20–25)[6,37].

The corresponding FEs and average yield rates were calculated according to Eqs. 1–5. The optimal applied potential for urea synthesis of CuWO₄ was the same as that of CuO (−0.2 V vs. RHE), which is more positive than that of WO₃ (−0.8 V vs. RHE) (Fig. 3b). The peak urea yield rate of 99.5 ± 3.0 μg h⁻¹ mg⁻¹ with a remarkable FE as high as 70.9 ± 2.2% could be obtained for the CuWO₄ catalyst at −0.2 V vs. RHE with a current density of nearly 1.0 mA cm⁻². By contrast, the maximum urea yield rate of CuO was 63.9 ± 2.2 μg h⁻¹ mg⁻¹ and the maximum FE of 25.4 ± 2.7%. However, for WO₃, a maximum urea yield rate of 61.7 ± 1.2 μg h⁻¹ mg⁻¹ and a urea FE of 24.7 ± 1.7% could be obtained only at a more negative operating potential (−0.8 V vs. RHE). The electrochemical active surface area (ECSA) was investigated and CuWO₄ still had the highest urea yield after normalization (Supplementary Figs. 26–27). To verify the accuracy of the results, the isotope ¹⁵N-labeled potassium nitrate (K¹⁵NO₃) was used as the electrolyte for CA measurements, and the yield rates of urea were quantified by ¹H NMR spectroscopy (Fig. 3c, Supplementary Figs. 28–29)[6]. The urea yield rate calibrated by NMR spectra was 98.5 ± 3.2 μg h⁻¹ mg⁻¹ with a urea FE of 70.1 ± 2.4% (Fig. 3d), similar to that measured using the urease method. Other control experiments were also supplemented, and no urea was detected in the absence of CO₂, NO₃⁻, catalyst, or potential (Supplementary Figs. 30–31). Therefore, the generated urea was a result of the coupling of the feeding CO₂ and NO₃⁻ via the electrocatalysis of CuWO₄; and no other C−N coupling products were detected.

Compared with the complex process of coupling CO₂ and NO₃⁻ to urea, side reactions such as NO₃⁻ reduction to NO₂⁻ or NH₃ and HER could occur more easily; as a result, the generation of by-products leads to low FEs of urea for most systems[18]. Therefore, investigating the FE of the entire NO₃⁻RR (Eq. 6) and the selectivity of products (Eq. 7) will help us understand the advantages of using CuWO₄ (Fig. 3e, f, Supplementary Fig. 21). At the low overpotential region (−0.1 to −0.2 V

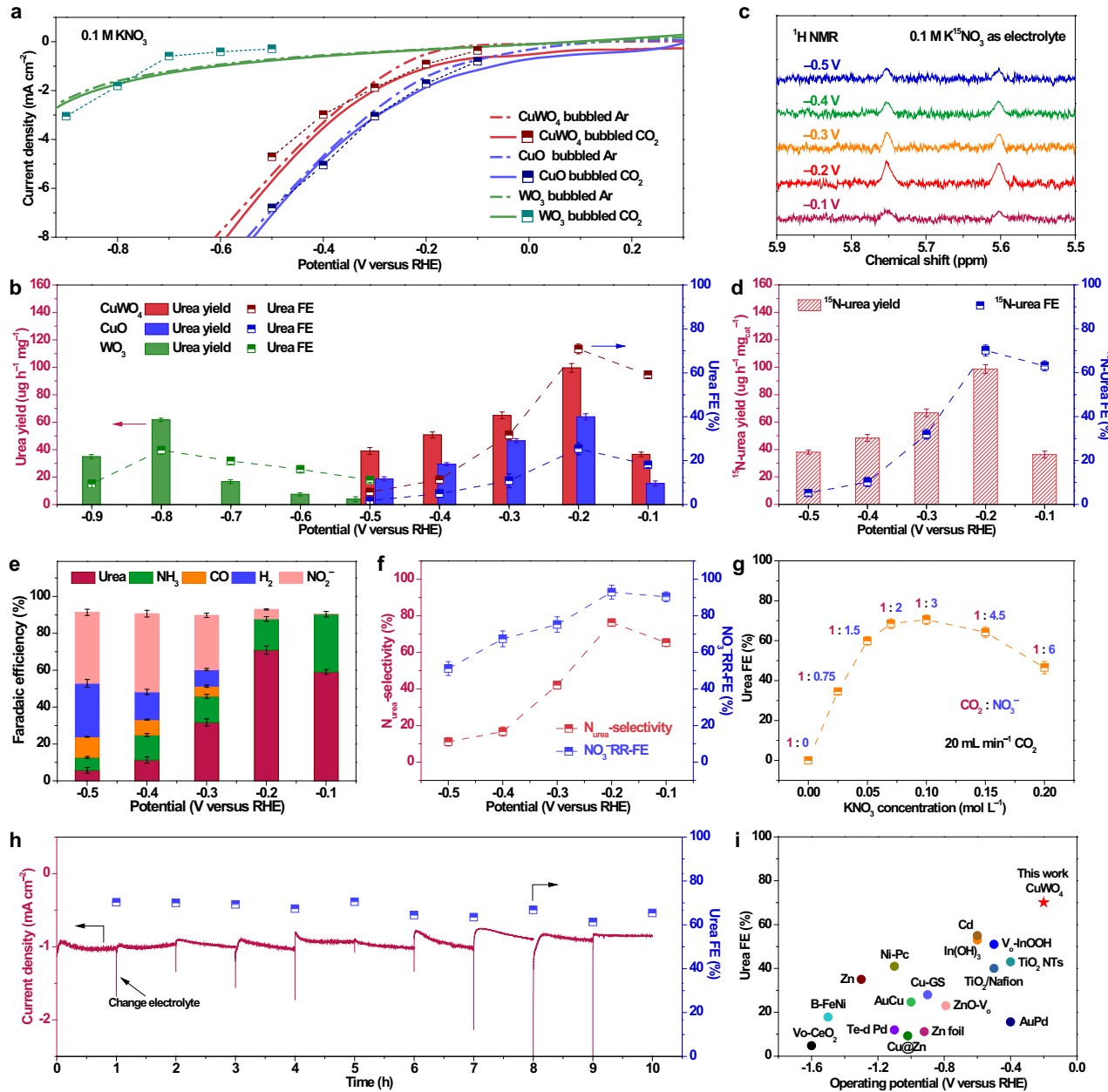

**Fig. 3 | Electrochemical synthesis of urea. a** LSV curves of $CuWO_4$, CuO, and $WO_3$ in 0.1 M $KNO_3$ with Ar or $CO_2$ bubbling and I–V plots of $CuWO_4$, CuO, and $WO_3$ in 0.1 M $KNO_3$ with $CO_2$ at different potentials. **b** Yield rates and FE values of urea production for $CuWO_4$ at different applied potentials in 0.1 M $KNO_3$ with $CO_2$ bubbling (20 mL min⁻¹). **c** ¹H NMR data of isotope calibration experiment in 0.1 M $K^{15}NO_3$ with $CO_2$ bubbling (20 mL min⁻¹) at different applied potentials. **d** ¹⁵N-urea yield rates and FEs via integrated peak area from NMR data. **e** FE values of all products for $CuWO_4$ at different applied potentials in 0.1 M $KNO_3$ with $CO_2$

bubbling (20 mL min⁻¹). **f** $N_{urea}$-selectivity and $NO_3^-$RR-FE for $CuWO_4$ at different applied potentials in 0.1 M $KNO_3$ with $CO_2$ bubbling (20 mL min⁻¹). **g** Urea FEs for $CuWO_4$ at −0.2 V vs. RHE in 0.1 M $KNO_3$ in different concentrations of $KNO_3$ electrolyte with $CO_2$ bubbling (20 mL min⁻¹). **h** Stability test of urea synthesis during 10 h of electrolysis at −0.2 V vs. RHE in 0.1 M $KNO_3$ with $CO_2$ bubbling (20 mL min⁻¹). **i** Comparison of the results of this work with state-of-art electrocatalytic synthesis urea catalysts in terms of operation potential and FE. **b**, **d**–**g** Error bars in accordance with the standard deviation of at least three independent measurements.

vs. RHE), the FEs of the entire $NO_3^-$RR were close to 100%, most of the $NO_3^-$ could react with $CO_2$ to form urea, and the nitrogen selectivity for urea reached 76.2 ± 1.7% at −0.2 V vs. RHE. Both the urea yield rate (Fig. 3d) and the nitrogen selectivity (Fig. 3f) gradually decreased under more negative applied potentials, where kinetically favorable competitive reactions occur. Significantly, the CuO could effectively catalyze the reaction of $NO_3^-$ to $NO_2^-$, which significantly decreased the FE of urea (Supplementary Figs. 16, 23); but $WO_3$ can effectively avoid the $NO_2^-$ generation (Supplementary Figs. 16, 25). With the assistance of high-valence W reaction center, the competitive reactions for urea synthesis catalyzed by $CuWO_4$ are effectively suppressed.

The ratio of two reactants may impact the efficiency and selectivity of the reaction. $CO_2$ with different flow rates was continuously fed, and the performance of $CuWO_4$ was measured at −0.2 V vs. RHE (Supplementary Fig. 32). When the flow rate of $CO_2$ was more than 5 mL min⁻¹, the urea FEs remained at approximately 70%; this is because when the flow rates are higher than the consumption rate, the $CO_2$ concentration remains roughly the same. The flow rate of $CO_2$ was fixed at 20 mL min⁻¹, and the concentration of $KNO_3$ in the electrolyte was changed. As shown in Fig. 3g and Supplementary Fig. 33, the urea FEs had a volcano peak-shaped distribution, reaching a peak at the $KNO_3$ concentration of 0.1 M. As the saturated concentration of $CO_2$

(0.033 M) in aqueous solution at standard temperature and pressure, the ratios of $CO_2$ and $NO_3^-$ in different concentrations of $KNO_3$ aqueous solution were 1:0, 1:0.75, 1:1.5, 1:2, 1:3, 1:4.5, and 1:6. Considering that the mass transfer efficiency of $CO_2$ may be higher owing to the small amount of bubble transport, the actual ratio of $CO_2$ and $NO_3^-$ in 0.1 M $KNO_3$ may be closer to the stoichiometric ratio of C and N atoms in urea, which is more conducive to a higher FE and the formation of urea. In addition, no other C−N coupling products (such as methylamine, ethylamine, formamide, acetamide, etc) were detected when the ratios of $CO_2$ and $NO_3^-$ changed.

The durability and reproducibility of $CuWO_4$ toward urea electrosynthesis were also examined. As shown in Fig. 3h, the $CuWO_4$/CP electrode maintained a relatively stable operation current density of approximately 1.0 mA cm$^{-2}$ for 10 h at −0.2 V vs. RHE with an average urea FE of 68.0%. The longer-term continuous electrolysis was also performed, and after 20 h of electrolysis without electrolyte renewal, the Faradaic efficiency of urea was 56.4% (Supplementary Figs. 34−35). The final Faradaic efficiency remained above 50% after three 20 h long-term repeated tests, demonstrating excellent reproducibility. After the reaction, no appreciable topography and crystal structure changes were observed for $CuWO_4$ (Supplementary Figs. 36−38). The only change detected is the binding energy of Cu on the surface of the $CuWO_4$ after electrolysis immediately transfer to the XPS under the protection of Ar (Supplementary Fig. 39)[38], indicating that the crystal structure of $CuWO_4$ is stable, and the lower valence state Cu involved during the electrosynthesis. Moreover. no evident vibrational changes of $CuWO_4$ (Supplementary Fig. 40) and $WO_3$ (Supplementary Fig. 41) were found in in situ Raman spectra at operating potentials. By contrast, for CuO, multiple vibration peaks corresponding to $Cu_2O$ and other Cu derivatives appeared after applying the working potentials (Supplementary Fig. 42)[39,40]. And the valence state of Cu in CuO did not change completely back to $Cu^{2+}$ after exposure to air, due to the structural transformation of CuO (Supplementary Figs. 42−43).

The FE of urea and the operating potential of this work were compared with those of previous reports (Fig. 3i and Supplementary Table 1)[41–45]; the FEs of urea in previous works were less than 53%, and the operating potentials were more negative than −0.4 V. The $CuWO_4$ catalyst system significantly improved the urea FE (70.1 ± 2.4%) and reduced the operating potential (−0.2 V vs. RHE) with a current density of nearly 1.0 mA cm$^{-2}$.

## Unraveling the origin of C−N bond formation and reaction mechanism

As $CuWO_4$ can obtain high FEs of urea at low operating potentials, an in-depth understanding of the C−N bond formation mechanisms of $NO_3^-$ and $CO_2$ co-reduction mechanism and structure−function relationship of $CuWO_4$ is attractive, which may be helpful for the development of more advanced catalytic systems.

In situ Raman spectroscopy (Supplementary Fig. 44 shows the setup) was used to probe the stretching vibrations of generated intermediates on the surface of the $CuWO_4$ under working conditions. Compared with the vibration peaks of $NO_3^-$ (1048 and 1365 cm$^{-1}$) under Ar-saturated conditions, new peaks at 1060 and 1351 cm$^{-1}$ corresponding to $*CO_3^{2-}$ and $*HCO_3^-$ species, respectively, emerged after $CO_2$ bubbling (Fig. 4a) at the open circuit state[46,47]. At the working state (potential range from 0.1 to −0.3 V vs. RHE), the peak at 1979 cm$^{-1}$ could be observed when the potential reached −0.1 V vs. RHE, which is assigned to the *CO on bridge sites[48]. Moreover, when the applied potential increased, the peak intensities of $*CO_3^{2-}$ and $*HCO_3^-$ gradually decreased, indicating that *CO came from these derivatives of $CO_2$. Interestingly, only one $NO_3^-$ reduction species could be detected at 1428 cm$^{-1}$ at the working state, corresponding to the $\upsilon(N{=}O)$ of $*NO_2$ intermediate in bridging configuration[49]. Under the potential of −0.2 V vs. RHE, the maximum peak intensities of 1428 and 1979 cm$^{-1}$ were reached, indicating that the highest coverage of $*NO_2$ and *CO

intermediates on the surface of $CuWO_4$ can be received. It is in agreement with the best-applied potential (−0.2 V vs. RHE) for the highest rate and FE of urea formation. As reaction intermediates irrelevant to the rate-determining step, and the product of rate-determining step could be rapidly consumed, which are difficult to be observed by non-time-resolved characterization methods. Thus, the undissociated intermediates that serve as substrates for the rate-determining step can be observed in Raman spectra[37]. Therefore, assuming that $*NO_2$ and *CO are the key intermediates in urea formation is reasonable, and the C−N bond formation between $*NO_2$ and *CO coupling is very likely the rate-determining step of the urea electrosynthesis reaction on the surface of $CuWO_4$. In contrast, various species from $NO_3^-$ and $CO_2$ reduction can be observed on the surface of the CuO (Supplementary Fig. 45). Stretching peaks of *CHO and $*CH_3O$ species at 1048 and 1110 cm$^{-1}$ are observed in the range from −0.3 to −0.5 V vs. RHE, which is consistent with the increased yield rate of methane (Supplementary Fig. 23)[50]. A variety of nitrogenous intermediates with weak peak intensities are observed, indicating that the coverage of nitrogenous intermediates was low, which are in agreement with the low yield rate and FEs of urea for CuO. The $WO_3$ exhibited a mismatch between the reduction potentials of $CO_2$ and $NO_3^-$ (Supplementary Fig. 46). The stretching peak at 2000 cm$^{-1}$, corresponding to the *CO on atop sites[48,51], could not be observed until the potential reached −0.6 V vs. RHE. As the applied potential changed, the intermediates of $NO_3^-$ reaction on the surface of $WO_3$ changed. In the range from −0.2 to −0.4 V vs. RHE, the main intermediate of $NO_3^-$ reduction was $*NO_2$, which is consistent with the observed for $CuWO_4$. When the applied potential increased from −0.5 to −0.7 V vs. RHE, the vibration peaks at 1143, 1163, 1390, and 1533 cm$^{-1}$ gradually appeared, corresponding to $\upsilon(N{-}O)$ and $\upsilon(N{=}O)$ of nitrito orientation, and $\upsilon(N{=}O)$ of nitroxyl for chelating nitrito[49]. These results explain the more negative potential requirement and low selectivity for $WO_3$, which is because the generation of *CO intermediate requires a relatively negative applied potential; at such a relatively negative potential range, the $NO_3^-$ reaction intermediate transformed from $*NO_2$ to other intermediates.

To further verify the speculations above, $NO_2$-temperature-programmed gas desorption (TPD) and CO-TPD were conducted to evaluate the thermodynamic adsorption energy of $*NO_2$ and *CO on the surface of CuO, $WO_3$, and $CuWO_4$. As shown in Fig. 4b, no evident $NO_2$ desorption peak was observed on the CuO $NO_2$-TPD curve, which means that the CuO surface hardly adsorbed $NO_2$. On the contrary, $WO_3$ showed a good adsorption capacity for $NO_2$ with a desorption peak in the low-temperature region (<200 °C) and multiple desorption peaks in the high-temperature region (200−400 °C), representing the physical and chemical adsorptions of $NO_2$ on the $WO_3$ surface, respectively[52]. The $NO_2$-TPD curve of $CuWO_4$ also shows a series of $NO_2$ desorption peaks. However, the strength is weaker than that of $WO_3$, indicating that $CuWO_4$ has a moderate adsorption capacity for $*NO_2$ intermediate by balancing the properties of CuO and $WO_3$. It is also consistent with the results of nitrite formation of the three catalysts (Supplementary Fig. 16). For the CO-TPD measurement (Supplementary Fig. 47), the peak strength of CO desorption for $WO_3$ in both low and high-temperature regions was higher, indicating that $WO_3$ had a stronger CO-adsorption capacity. CuO also showed a certain degree of chemical adsorption for CO. With the influence of octahedral $[WO_6]$ clusters, the physical adsorption of CO by $CuWO_4$ significantly increased; however, the CO chemical adsorption capability of $CuWO_4$ was the same as that of CuO, suggesting that $CuWO_4$ maintains the advantages of higher physical adsorption ability of the W sites, and the favorable chemical adsorption capability of Cu sites for the *CO intermediate. Therefore, as predicted, $CuWO_4$ could provide appropriate adsorption and stabilization capability for both $*NO_2$ and *CO intermediates compared to $WO_3$ and CuO, which is consistent with the in situ Raman spectroscopy results.

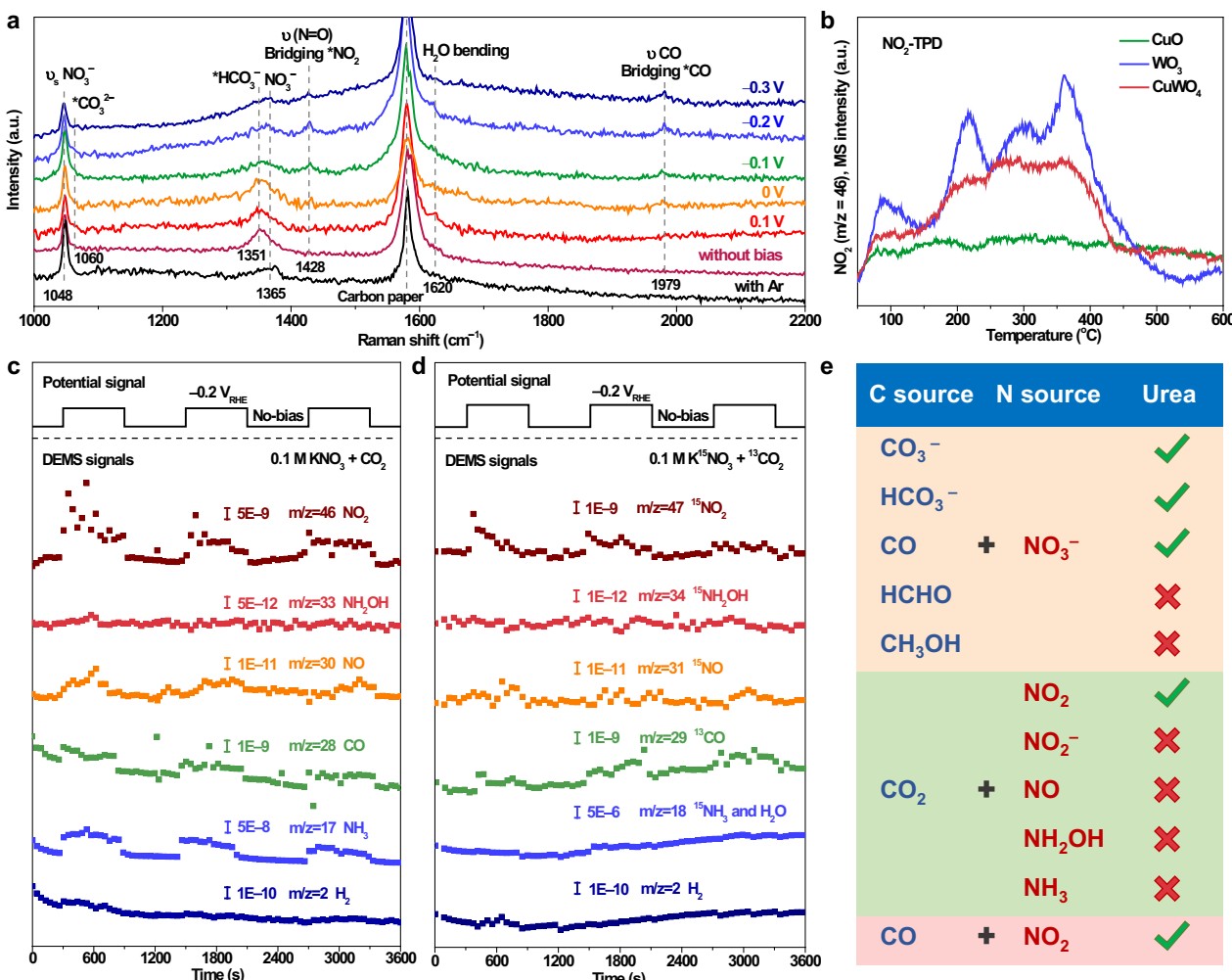

**Fig. 4 | In-situ characterization and inference of intermediate species. a** In situ Raman spectra of $CuWO_4$ in 0.1 M $KNO_3$ with $CO_2$ bubbling at different applied potentials or Ar bubbling at open circuit state. **b** $NO_2$-temperature programmed gas desorption (TPD) of $CuWO_4$, CuO, and $WO_3$. **c** Online DEMS of $CuWO_4$ in 0.1 M $KNO_3$ with saturated $CO_2$ at −0.2 V vs. RHE. **d** Online DEMS of $CuWO_4$ in 0.1 M $K^{15}NO_3$ with saturated $^{13}CO_2$ at −0.2 V vs. RHE. **e** Control experiment results of different carbon and nitrogen sources for speculating the urea synthesis mechanism.

Differential electrochemical mass spectrometry (DEMS) can detect the products of the dissociated electrochemical-generated intermediates on the surface of the catalyst (Supplementary Fig. 48)[38]. The mass-to-charge ratio (m/z) signals of 2, 17, 28, 30, 33, and 46 corresponding to $H_2$, $NH_3$, CO, NO, $NH_2OH$, and $NO_2$, respectively, were recorded (Fig. 4c). For $CuWO_4$, the signals of $NH_3$ (m/z = 17), CO (m/z = 28), NO (m/z = 30), and $NO_2$ (m/z = 46) showed fluctuations consistent with the switching cycles of open circuit and working states (at −0.2 V vs. RHE). The high captured signal strengths of $NH_3$ (m/z = 17) corroborate that $NH_3$ is the main by-product for $CuWO_4$ at the applied potential of −0.2 V vs. RHE. Considering that a small amount NO may come from the disproportionation reaction of $NO_2$ and the reaction during $NH_3$ production, the observation of $NO_2$ and CO, the dissociated electrochemical-generated intermediates, suggests that $*NO_2$ and $*CO$ are the critical intermediates for the C–N bond formation on the $CuWO_4$ surface, which also corroborates the in situ Raman measurements. To confirm these results, the reactants $KNO_3$ and $CO_2$ were replaced with $K^{15}NO_3$ and $^{13}CO_2$. As shown in Fig. 4d, corresponding isotopic molecule signals of $^{13}CO$ (m/z = 29), $^{15}NO$ (m/z = 31), and $^{15}NO_2$ (m/z = 47) could be detected, indicating that $*CO$ and $*NO_2$ intermediates were generated from the reduction of $CO_2$ and $NO_3^-$ rather than pollutants. The signals of $NH_3$ (m/z = 17) and $NO_2$ (m/z = 46) could be detected for $WO_3$ (Supplementary Fig. 49a); however, CO (m/z = 28) could not be clearly observed, which confirms that $WO_3$ can only reduce $NO_3^-$ to produce $NH_3$ rather than trigger urea generation at −0.2 V vs. RHE, and that $*CO$ generation potential is mismatched with that of $NO_2$ formation (Supplementary Fig. 46). For CuO (Supplementary Fig. 49b), relatively high signal strengths of $NO_2$ (m/z = 46) and CO (m/z = 28) could be detected, which is identical to the results that urea generation could occur at −0.2 V vs. RHE. However, these signals displayed a pulse-like intensity decay, which may be attributed to the weaker adsorption capacity of intermediates or insufficient structural stability of CuO. At the same time, considerable CO, $H_2$, and $NH_3$ were observed, which is consistent with the lower selectivity of urea electrosynthesis for CuO (Supplementary Fig. 23c).

The C- or N-sources were further changed to screen the C–N coupling process. Derivatives of $CO_2$ in the aqueous phase ($CO_3^-$ and $HCO_3^-$) and the desorption species (CO, HCHO, and $CH_3OH$) of the intermediates generated during the $CO_2$ reduction were used as the C-source for urea electrosynthesis with $NO_3^-$. As shown in Fig. 4e and Supplementary Table 2, $CO_3^-$, $HCO_3^-$, and CO can be co-reactants with $NO_3^-$ for urea electrosynthesis by $CuWO_4$; HCHO and $CH_3OH$, the deeper $CO_2$ reduction products, cannot conduct the process for $CuWO_4$; which confirms that the $*CO$ should be the key $CO_2$ reduction intermediate on $CuWO_4$ before C–N coupling. By contrast, $NO_2$, $NO_2^-$, NO, $NH_2OH$, and $NH_3$, possible desorption species of the

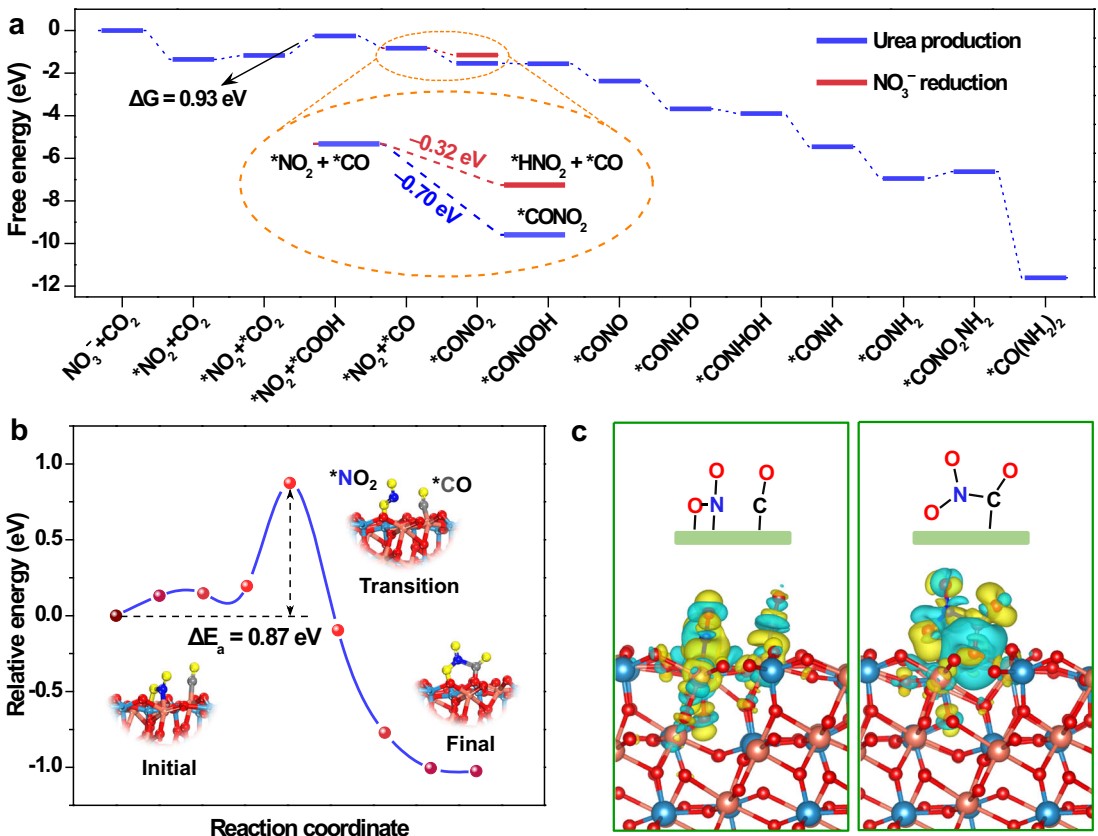

**Fig. 5 | DFT calculation of urea synthesis mechanism on bimetallic CuWO$_4$ (111) surface. a** Free-energy diagram for urea production and NO$_3^-$ reduction on the CuWO$_4$ (111) facet. **b** Mechanism of C−N coupling between *CO and *NO$_2$. The initial, transition, and final states during the *CONO$_2$ formation are presented. Gray, blue, red, orange, cyan, and yellow balls represent C, N, lattice O, Cu, W, and adsorbate O, respectively. **c** Charge density difference of the co-adsorption state of *CO and *NO$_2$ (left side), and adsorbate *CONO$_2$ (right side) on the CuWO$_4$ (111) facet. The iso-value is 0.002 e/Å$^3$. Electron accumulation and depletion are represented by yellow and cyan sections, respectively.

intermediates for NO$_3^-$RR, were used as the N-source for urea electrosynthesis with CO$_2$. The results show that only NO$_2$ can serve as the co-reactant with CO$_2$ to form urea for CuWO$_4$, inferring that *NO$_2$ should be the critical NO$_3^-$ reduction intermediate in the process of C−N coupling for urea generation. Electrosynthesis of urea on the surface of CuWO$_4$ could also occur when CO and NO$_2$ were used as feedstocks, further proving that *CO and *NO$_2$ are the critical intermediates in the reaction pathway of urea electrosynthesis on the CuWO$_4$ surface.

To further unravel the fundamental origins of the high selectivity of CuWO$_4$, density functional theory (DFT) calculations (Methods) were conducted on the preferentially oriented in the (111) direction of CuWO$_4$ crystal (Supplementary Fig. 50) corresponding to the XRD and TEM results. Figure 5a and Supplementary Fig. 51 show the detailed free-energy diagram of the corresponding structures with the lowest energy pathway from CO$_2$ and NO$_3^-$ to urea. The reaction started with the reduction of NO$_3^-$ to *NO$_2$ with strong adsorption-free energy of −1.35 eV. By contrast, CO$_2$ was first physically adsorbed on the surface of CuWO$_4$. The hydrogenation of *CO$_2$ is a potential determining step with a 0.93 eV uphill in reaction-free energy; whereafter, the *COOH was spontaneously reduced to *CO thermodynamically. As shown in Supplementary Fig. 51, in the relevant atomic configurations with the lowest energy, the *NO$_2$ intermediate was adsorbed as the bridging nitro with the N bonding to Cu and O bonding to W, suggesting that the W sites play an important role in stabilizing *NO$_2$. Whereas, the *CO intermediate was located at the bridge site between two Cu atoms, consistent with the in situ Raman results. As the *CO was formed,

the *NO$_2$ intermediate was involved in the urea production. In contrast to the reaction-free energy for hydrogenation of *NO$_2$ to *HNO$_2$ (−0.32 eV), the lower reaction-free energy of −0.70 eV for *CONO$_2$ formation (insert in Fig. 5a) on the surface of CuWO$_4$ was beneficial for the direct C−N coupling at a very early stage. Simultaneously, the reaction-free energies of *CO to *CHO (0.17 eV) and *CO to *COH (2.12 eV) are higher compared to the C−N coupling process (−0.70 eV, Supplementary Fig. 52). As a result, the NH$_3$-related or *CO hydrogenation side reactions could be significantly suppressed, which may explain the high selectivity of urea production by CuWO$_4$. Although the free energy of C−N bonding between *NO$_2$ and *CO is lower thermodynamically, an activation energy barrier as high as 0.87 eV for the *CONO$_2$ formation process exists (Fig. 5b), making the coupling of *CO and *NO$_2$ a slow kinetics process compared to the other elementary reactions during urea production, which is in agreement with the experimental results that the C−N bond formation between *NO$_2$ and *CO is the rate-determining step. Further calculations of the charge density difference of *NO$_2$ and *CO adsorbed states on the (111) facet of CuWO$_4$ were investigated (Supplementary Fig. 53). The electrons on the CuWO$_4$ could transfer to the *CO and *NO$_2$, corresponding to the nature of the reduction reaction, which enables the improvement of their intrinsic activity and realizes the C−N coupling between the two intermediates[16]. Compared with the change of charge region before and after C−N bonding (Fig. 5c) an evident electron exchange between the *CO and *NO$_2$ intermediates was observed; the electrons mainly flowed from the C atom of *CO to the N atom of *NO$_2$, and completed the C−N coupling. The further hydrogenation of

*CONO$_2$ to *CONH$_2$ intermediate can occur in an energetically favorable pathway, which has an advantage over the formation of *CONONO$_2$, *CONHONO$_2$, and *CONHNO$_2$ in the reaction-free energy (Supplementary Fig. 54). Although the free energy of the coupling of *CONH$_2$ with the second *NO$_2$ increased slightly, it was not sufficient to prevent a thermodynamically spontaneous pathway of urea formation. These results theoretically confirm that the rate-determining step is the C−N coupling between the *CO and *NO$_2$ intermediates on the surface of CuWO$_4$ for urea electrosynthesis. Furthermore, the lower reaction-free energy of the C−N coupling between the *CO and *NO$_2$ intermediates and the thermodynamically spontaneous formation pathway determine the extremely high selectivity of urea electrosynthesis on the surface of CuWO$_4$, in line with the preceding experimental analysis.

## Discussion

Inspired by the efficient nitrate assimilation process by the high-valence Mo-based reaction center of nitrate reductase in nature, CuWO$_4$ catalyst with the high-valence W reaction centers was designed and employed for urea electrosynthesis with CO$_2$ and NO$_3^-$ as the feedstocks. The prepared CuWO$_4$ could achieve highly efficient urea production with a $98.5 \pm 3.2 \, \mu g \, h^{-1} \, mg^{-1}_{cat}$ urea yield at −0.2 V vs. RHE and high FE of $70.1 \pm 2.4\%$. As evidenced by in situ Raman, TPD, DEMS, C/N-sources changing experiments, and theoretical calculation, early C−N coupling originating for urea electrosynthesis from *NO$_2$ and *CO intermediates on the surface of CuWO$_4$ was verified. Owing to the relative positive formation potential and appropriate adsorption ability of *NO$_2$ and *CO intermediates on the alternating bimetallic W and Cu sites, CuWO$_4$ optimized the coupling for these critical intermediate species. The lower energy barrier of direct coupling of *NO$_2$ and *CO enabled the C−N coupling to easily occur. As primary intermediates of NO$_3^-$ and CO$_2$ reduction, the coupling of *NO$_2$ and *CO decreased the possibility of intermediates desorption from complex elementary reactions before the C−N formation, which inhibits other side reactions. This study may also provide new inspiration for designing electrochemical synthesis systems to produce a broader range of high-value C−N bond compounds.

## Methods

### Preparation of catalysts

CuWO$_4$ nanoparticles were synthesized by the hydrothermal method. Na$_2$WO$_4$·2H$_2$O (0.33 g) and CuSO$_4$ (0.16 g) were dissolved in 20 mL of ultrapure water, respectively. In sequence, the Na$_2$WO$_4$·solution was slowly dripped into the solution of CuSO$_4$ under vigorous stirring. The mixture was transferred to a Teflon-lined stainless steel autoclave (50 mL), and maintained at 180 °C for 20 h in an oven. For CuO nanoparticles, 20 mL of 0.5 M Na$_2$CO$_3$ was slowly dripped into CuSO$_4$ solution (0.16 g in 20 mL H$_2$O) under vigorous stirring. For WO$_3$ nanoparticles, HCl solution (0.1 M, 20 mL) was slowly dripped into the Na$_2$WO$_4$·solution (0.33 g in 20 mL H$_2$O) under vigorous stirring. The hydrothermal synthesis procedures were the same as that of CuWO$_4$. After hydrothermal synthesis, all samples were separated and washed by centrifugation. Ultimately, all samples were calcined at 500 °C in the air for 2 h to remove organic pollutants and improve the crystallinity.

### Electrode preparation and electrochemical measurements

All electrochemical measurements were conducted on a CHI760E electrochemical instrument. A mass loading of 1 mg cm$^{-2}$ was used on carbon paper (CP) working electrodes in a three-electrode H-cell electrolyzer, at room temperature and atmospheric pressure. 0.01 g of catalyst powders and 50 uL of Nafion solution were dissolved in 1 mL of ethanol as the ink. Subsequently, catalyst ink (100 μL) was coated on carbon paper (Toray 060) (geometric area 1 × 1 cm$^2$), and the working

electrode was obtained after drying (catalyst loading of 1 mg cm$^{-2}$). A Nafion 117 membrane (Dupont) was used for separation. A ruthenium oxide-coated titanium sheet was used as the counter electrode. A leaking-free Ag/AgCl (saturated KCl) electrode was used as the reference electrode. KNO$_3$ solutions (0.1 M as N-source) with or without high-purity CO$_2$ (20 mL min$^{-1}$ with bubbling, as C-source) were employed as the optimized electrolytes for the electrochemical measurements. Before tests, the corresponding gas was used to pre-saturate the electrolyte of the cathode part. In the electrolytic process, the gas flow rate was set as 20 mL min$^{-1}$. The LSV measurements were performed with a negative scan direction and a scan speed of 10 mV/s. For each experiment, the applied potential was converted to scales of RHE according to $E_{(vs.RHE)} = E_{(vs.Ag/AgCl)} + E_{Ag/AgClvs.RHE}$, where $E_{Ag/AgClvs.RHE}$ is the potential difference between the Ag/AgCl electrode and a commercialized RHE (HydroFlex®) under corresponding conditions (such as saturated Ar or CO$_2$). The same methods were adopted for isotope-labeling experiments and operando Raman measurements.

### Quantitative analysis and identification of urea, ammonia, nitrite, and hydrazine

The content of urea, ammonia, and hydrazine was determined by urease, diacetylmonooxime, indophenol blue, and Watt and Christo, respectively[6,15]. For the urease decomposition method, 0.5 mL of urease solution (urease: 5 mg mL$^{-1}$) was added into 4.5 mL of electrolyte, and then reacted at 40 °C for 40 min. The urease solution also contained 0.1 g ethylenediaminetetraacetic acid disodium salt and 0.49 g K$_2$HPO$_4$ per 100 mL. Then, the NH$_3$ concentrations of electrolyte before and after decomposition were detected by ion chromatograph and indophenol blue method. The urea yield was calculated according to the NH$_3$ concentrations before and after decomposition. For the diacetylmonoxime method, two chromogenic solutions, A and B, were prepared. For solution A, 10 mL of phosphoric acid, 30 mL of concentrated sulfuric acid, and 10 mg of ferric chloride were added to 60 mL of deionized water; finally, deionized water was added such that the volume reached 100 mL. For solution B, 0.5 g of diacetylmonoxime and 10 mg of thiosemi-carbazide were dissolved in 100 mL of deionized water. 1 mL of the post-test electrolyte was mixed with 2 mL of solution A and 1 mL of solution B and heated at 100 °C for 15 min. After cooling, the absorbance of the solution at 525 nm was measured by a UV-vis spectrophotometer (Thermo One Plus). The calibration was performed using a standard concentration-absorbance curve of urea solution, including urea concentrations in the electrocatalytic test (Supplementary Fig. 12). For the indophenol blue method, 10 ml of electrolyte was added to 2 mL of salicylic acid solution (5 wt%), 1 mL of sodium hypochlorite solution (0.05 mol L$^{-1}$), and 0.2 mL of sodium nitrosoferricyanide solution (1 wt%). After uniform mixing, the sample solution was placed for 45 min and tested with a UV-vis spectrophotometer (at 655 nm). The calibration was performed using a standard concentration-absorbance curve of ammonia solution, including ammonia concentrations in the electrocatalytic test (Supplementary Fig. 18). The concentration of ammonium ion in the electrolyte was also detected by ion chromatography (Thermo Scientific Dionex Aquion). The calibration was performed using a standard chromatographic curve of ammonium solution, including ammonium concentrations in the electrocatalytic test (Supplementary Fig. 19). For Watt and Christo's method, 5 mL of a reagent consisting of a mixture of p-dimethylaminobenzaldehyde (5.99 g), concentrated hydrochloric acid (30 mL), and ethanol (300 mL) was added to 5 mL of electrolyte. The absorbance of the resulting solution was measured at 455 nm. The concentration of nitrite ions in the electrolyte was detected by ion chromatography (Thermo Scientific Dionex Aquion). The calibration was performed using a standard chromatographic curve of nitrite solution, including nitrite concentrations in the electrocatalytic test (Supplementary Fig. 14).

The FE is the ratio of the number of electrons transferred between the formation of products and the total current flowing through the circuit. As 16 electrons are required to form a urea molecule, and 8 electrons are required to form an $NH_3$ molecule, the FE of urea and $NH_3$ can be calculated, respectively, as follows:

$$FE_{urea}(\%) = (16 \times F \times c_{urea} \times V)/(60.06(g/mol) \times Q) \times 100\%, \quad (1)$$

$$FE_{NH_3}(\%) = (8 \times F \times c_{NH_3} \times V)/(17(g/mol) \times Q) \times 100\%, \quad (2)$$

$$FE_{NO_2^-}(\%) = (2 \times F \times c_{NO_2^-} \times V)/(46(g/mol) \times Q) \times 100\%, \quad (3)$$

where $c_{urea}$ and $c_{NH_3}$ (µg/mL) are the measured urea and $NH_3$ concentrations, respectively; $V$ (mL) is the total volume of the electrolyte, F is the Faraday constant (96,485.3 C mol$^{-1}$); and $Q$ (C) is the total charge passed through the working electrode.

The average yield rates of urea and $NH_3$ were calculated according to the following equation.

$$R_{urea} = (c_{urea} \times V)/(t \times m), \quad (4)$$

$$R_{NH_3} = (c_{NH_3} \times V)/(t \times m), \quad (5)$$

where $t$ is the time (h) for electrocatalysis and $m$ is the catalyst loading (mg).

The $NO_3^-$-RR-FE was calculated as follows:

$$FE_{NO_3^- RR}(\%) = \frac{Q_{NO_3^- RR}}{Q} \times 100\%, \quad (6)$$

where $Q_{NO_3^- RR}$ represents the charge consumed by the products involved in nitrate reduction.

The $N_{urea-selectivity}$ was calculated as follows:

$$N_{urea-selectivity} = n_{urea}/n_{total}, \quad (7)$$

where $n_{urea}$ is the number of moles of nitrogen in as-produced urea, and $n_{total}$ is the total number of moles of N atoms in the products from $NO_3^-$-RR.

### Isotope-labeling product quantification and identification
The isotope-labeling experiments for the identification of products were conducted using a solution of 0.1 M K$^{15}$NO$_3$ as the electrolyte. The concentrations of $^{15}$N-urea and $^{15}$N-ammonia were both quantified by NMR[6,37]. For NMR sample preparation, the test solution was mixed with dimethyl sulfoxide at a ratio of 9:1 and shaken evenly. The NMR solution used to detect ammonia was adjusted to a pH of 3 by adding an appropriate amount of 1.0 M HCl solution. The NMR test was conducted under the water suppression mode with a scanning circle of 420 times. $^{15}$N-urea and $^{15}$NH$_4$Cl were used to prepare the standard solution, and the standard linear curves between the NMR signal and the product concentration were established (Supplementary Fig. 28). The yield rates and FEs were calculated by reference to the calculation formula in the spectrophotometer method.

### Quantitative analysis of other gas and liquid-phase products
The yield rates of CO, $CH_4$, $H_2$, and $N_2$ in gas products were determined by an online gas chromatograph (GC-2014, Shimadzu) with thermal conductivity and flame ionization detectors. Other liquid-phase products were identified by $^1$H NMR.

### $NO_2$ and CO temperature program desorption
The $NO_2$- and CO-TPD measurements were completed by AutoChem 2920 temperature program instrument and Hiden QIC-20 mass spectrometer. First, the catalyst was pretreated, and the temperature was raised to 200 °C with Ar flow for 0.5 h. In sequence, the catalyst was cooled down to 25 °C and exposed to $NO_2$ or CO flow of 30 mL min$^{-1}$ for 30 min. Before desorption, the sample was flushed in Ar gas for 10 min. Subsequently, $NO_2$ or CO desorption was performed in the range of 25–600 °C at a heating rate of 10 °C min$^{-1}$ under an Ar flow of 30 mL min$^{-1}$. The mass-to-charge ratio (m/z) signals of 28, 44, and 46 corresponding to CO, $CO_2$, and $NO_2$, respectively, were recorded.

### In-situ Raman spectroscopy measurements
In-situ Raman spectra were obtained from a homemade H-type three-electrode spectroelectrochemical cell with an embedded quartz window under the same working conditions of activity measurements (Supplementary Fig. 44 shows the experimental setup). The electrochemical workstation provided the corresponding applied potential. Before electrolysis, Ar or $CO_2$ was injected into the electrolyte for 20 mins to purge, and the gas flow rate was maintained at 20 mL min$^{-1}$ during the electrolysis. The Raman data was recorded after 15 mins of electrolysis to ensure that enough intermediates had accumulated on the surface of the catalyst. The laser intensity, the data recording circle, and the sweep speed were kept consistent.

### XPS measurement after electrolysis
The $CuWO_4$ and CuO electrodes were electrolyzed for an hour under at −0.2 V vs. RHE in 0.1 M $KNO_3$ with $CO_2$ bubbling (20 mL min$^{-1}$). After the test, the electrodes were treated instantly with deionized water and vacuum drying (20 °C). The Ar-protected samples were transferred to an Ar-filled sealed chamber for storage. The samples, which were not protected by Ar, were exposed to air for an hour. Then, the two types of samples were tested together by XPS.

### DEMS measurements
Differential electrochemical mass spectrometry (DEMS) test employed a high precision three-stage filter quadrupole mass analyzer with softer ionization (Ionic energy: 4–150 eV) (Supplementary Fig. 48a). Hiden QMS has a unique soft ionization technology. By optimizing the tuning of the electron energy of the EI source, the fragmentation peak can be reduced and the prominent molecular ion peak can be strengthened, which achieves the purpose of reducing interference. The DEMS cell was a dual thin-layer flow cell customized by Hiden, which was reported in previous work (Supplementary Fig. 48b)[53]. The working electrode was a custom-made glass carbon electrode uniformly coated with 50 µL catalyst ink (in Electrochemical measurements). The counter electrode was a ruthenium oxide electrode. Nafion 117 membrane was used as the separation membrane. The reference electrode was an Ag/AgCl electrode calibrated by a commercial RHE (HydroFlex). The electrolyte was 0.1 M $KNO_3$ solution, and the cathode electrolyte was saturated with $CO_2$. A microinjector ensured electrolyte flow in the DEMS cell at a flow rate of 200 µL min$^{-1}$. The electrochemical workstation provided the corresponding applied potential. In isotope-labeling experiments, 0.1 M K$^{15}$NO$_3$ and $^{13}$CO$_2$ were used as feedstocks.

### DFT calculations
All the DFT calculations in this work were conducted using the Vienna ab initio simulation program (VASP)[54,55]. The core-valence interactions were described by the projector-augmented wave (PAW) method[56,57], and the cut-off energy of plane-wave basis expansion was set to 450 eV. All spin-polarized calculations were performed by the generalized gradient approximation (GGA) and Perdew–Burke–Ernzerhof (PBE) for the exchange and association of functions[58]. The lattice constants were calculated to be $a = 4.681$ Å, $b = 5.867$ Å, and $c = 4.898$ Å, consistent with previous results, including calculated constants and experiment

values[58–61]. For the surface construction, the $CuWO_4$ surface was cleaved along a (111) direction, which is the dominant surface observed in XRD. A p(2 × 2) $CuWO_4$ (111) surface with six atomic layers was modeled. The two atomic layers at the bottom remain fixed to mimic the bulk phase, while the other layers were fully relaxed. A -15 Å vacuum layer was employed to eliminate the interaction of adjacent slabs. The free molecules of $HNO_3$, $H_2O$, $H_2$, and $CO_2$ were placed in a $(15 \times 15 \times 15)$ Å$^3$ cubic box to diminish the interplay between neighboring molecules. A $2 \times 2 \times 1$ Monkhorst–Pack k-point mesh sampling was utilized for all optimizations. When the forces on the relaxed atoms became less than 0.05 eV/Å, and the energies in the self-consistent iterations reached $10^{-5}$ eV, the optimized structures converged. The van der Waal (vdW) interaction was described using the DFT-D3 method[62,63]. The climbing image nudged elastic band (CI-NEB) method was used for locating the transition states (TSs)[64,65].

The Gibbs free energy can be expressed as:

$$\triangle G = \triangle E + \triangle ZPE - T \times \triangle S, \tag{8}$$

where $\triangle E$ is the reaction energy calculated by the DFT methods. $\triangle ZPE$ and $T \cdot \triangle S$ are the thermodynamic corrections of zero-point-energy (ZPE) and entropy (S) derived from the vibrational partition function at 298.15 K, respectively. Gaussian 03 software package was used for gas-phase species to calculate the thermodynamic corrections for the ideal gas approximation.

### Reporting summary
Further information on research design is available in the Nature Portfolio Reporting Summary linked to this article.

## Data availability
All data that support the findings of this study are present in the paper and the Supplementary Information. Additional data related to the study are available from the corresponding author upon reasonable request.

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

## Acknowledgements

This work was conducted by the Fundamental Research Center of Artificial Photosynthesis (FReCAP), financially supported by the National Key R&D Program of China (2022YFA0911904), the National Natural Science Foundation of China (NSFC) (Grant nos. 22172011 and 22088102), the Fundamental Research Funds for the Central Universities (DUT22LK06 and DUT22QN213), the Key Laboratory of Bio-based Chemicals of Liaoning Province of China, and the starting grant of Westlake University and the Kunpeng program of Zhejiang province.

## Author contributions

Y.Z., L.S., and Fu.L. conceived the project design and initiated the project. Y.Z. performed catalyst synthesis, most of the structural characterization, and electrochemical measurements. All spectra and electrochemical kinetics data were analyzed and interpreted by Y.Z. and Fu.L. Y.D. performed DFT calculations. W.L. and C.L. performed operando Raman spectra measurements and data analysis. Y.L. performed the SEM and TEM measurements. Z.Z. and Fe.L. performed partial product analysis of electrochemical measurements. Y.S. performed the XPS measurements and data analysis. All authors contributed to the discussions. Y.Z. wrote the paper with inputs from the other authors, and

Fu.L., Fe.L., and L.S. revised the manuscript. All authors reviewed the paper. Fu.L. supervised the research.

## Competing interests

The authors declare no competing interests.
