## [Peer Review File · Nature Communications]

REVIEWER COMMENTS

Reviewer #1 (Remarks to the Author):

In this manuscript, a CuWO₄ catalyst is used to synthesis urea from carbon dioxide and nitrate. The Faraday efficiency is 70.1% under a low operating potential (-0.2 V vs. RHE) and the urea production rate is high. The catalysts are well characterised and the possible reaction mechanism is explored. The manuscript should address the following question before it could be published.

- 1) From XRD results, there are other facets besides (111) facet. How these facets contribute to electrocatalysis processes?
- 2) The valence state of Cu is +2 in CuWO₄ while the lower valence state Cu involved during the electro synthesis (page 10). How the lower valence state Cu contribute to electrocatalysis processes?
- 3) In Figure 5a, *NO₂ + *CO could form *CONO₂, or *HNO₂ + *CO. Here, *CO could also be reduced. What's the free energy if only *CO is reduced while *NO₂ remains?
- 4) When the first C-N bond formed, it's energy favoured. When the second C-N bond formed, it's energy unfavoured. Should the second C-N bond form with other intermediates, such as CONO₂, CONO, CONHO?
- 5) The free energy increases 0.93 eV from *NO₂ + *CO₂ to *NO₂ + *COOH, which is much larger than that of C-N bond formation between *NO₂ and *CO. Why the latter is the rate-determining step?
- 6) The lattice constants were calculated to be a = 4.681 angstrom, b = 5.867 angstrom, and c = 4.898 angstrom. Are these data from optimised structure?
- 7) It's hard to see Cu in Figure S46 and Figure S47.

Reviewer #2 (Remarks to the Author):

The authors prepare a CuWO₄ material which they use for simultaneous electroreduction of nitrate and CO₂, suggesting that adsorbed intermediates recombine to form urea, still with low current density but relatively high faradaic efficiency. Even though I have concerns on how this system can lead to urea formation (comment 1 below), the authors do provide experimental evidence. Therefore, the manuscript could be published in Nature Communications, but some comments should be addressed first:

1. The authors used an unbuffered electrolyte, so the alkaline interfacial environment that is created due to the H₂O, CO₂ and NO₃⁻ reduction reactions is likely turning CO₂ at the interface to carbonate, which is not reactive. Therefore, it is unclear how CO₂ is eventually present at the interface to be reduced to CO. In this electrolyte, I suspect that even the solution pH has increased. The weak buffer that is created using CO₂ gas, or the stirring of the electrolyte (as it looks from the image) will not be sufficient. I understand that the authors provide experimental data that support urea is formed, but on the contrary the hydroxide formation during the above reduction reactions and the CO₂/bicarbonate/carbonate equilibria are unambiguous facts. The authors should very carefully consider the above and provide a clear explanation of their view of the interfacial conditions.
2. Elaborating a bit more on the choice of electrolyte, the authors state that the KNO₃ electrolyte was the "optimized electrolyte". Please explain what you mean, what was it compared with? Note that since there is no supporting electrolyte, just KNO₃ which is reacting, its concentration is decreasing

with time. A solid justification of the choice of electrolyte is needed, because it looks like the experiments were performed at conditions (nitrate concentration and pH) that were changing continuously.

3. The authors' hypothesis is that they combine sites that reduce NO_3^- to NO_2 and other that reduce CO_2 to CO , and thereby they facilitate the NO_2+CO recombination, which they believe are the critical educts to form urea. I like the fact that the authors were driven by a hypothesis, but:

(a) The authors say if there were "continuous copper sites" this would lead to hydrocarbon formation. Are the authors sure that copper in their material is truly isolated? How can they confirm they don't have adjacent copper atoms that will carry out the further reduction of CO ?

(b) Copper is good for reducing nitrate to ammonia, but the whole analysis considers only tungsten responsible for nitrate reduction.

(c) It is unclear why the authors didn't use another material combination to support their arguments. For example, would it be possible to use silver instead of copper?

4. How do the authors explain the higher current for CuO vs the CuWO_4 from the voltammetry? This contradicts with their interpretations.

5. DEMS is capable of detecting volatiles only. Therefore, the ammonia species detected in DEMS is different than those detected in the electrolyte with other methods. In addition, the authors' statement that NH_3 is detected in DEMS confirms that the solution is becoming alkaline, otherwise volatile ammonia would have not been formed in detectable amounts. Please consider both points and comment accordingly in the manuscript.

6. Why did the authors try to monitor CO at m/z 28? Given that the solution is saturated in CO_2 , significant fragmentation will occur in the EI and will lead to a very high background for CO . Did the authors use 70 eV or softer ionization?

7. A last comment on the concept description in first paragraphs of the intro. I think it gives an incomplete picture; the urea market is globally very large and the process described by the authors would require enormous amounts of feedstocks, i.e. CO_2 and nitrate. Regarding CO_2 , the intro now neglects that industrial urea synthesis is integrated with ammonia synthesis with grey hydrogen, so there is a point source of CO_2 . Regarding nitrate, (which is currently made by ammonia) the authors state that it can originate from wastewater (but concentrations are very low) or plasma oxidation of N_2 (which is however energy intensive). A bit more balanced discussion here on the scale challenges would be more accurate.

Reviewer #3 (Remarks to the Author):

The manuscript by Zhao et al. presents a clear hypothesis on the design of an electrocatalysts for the conversion of nitrate and carbon dioxide to urea (Figure 1 is great) and confirm it with an impressive set of experimental and computational data. The mechanistic studies presented by the authors might be of particularly high value to the field.

In principle, the manuscript might be suitable for publication in Nature Communications, if the authors can address the key comments on the stability of the CuWO_4 catalyst and urea analysis. Otherwise, the paper should be published in a more specialised journal upon addressing the comments below.

1. Why was it necessary to change the solution after every 1 h of electrolysis in Fig. 3h? The only reason the reviewer could think of is a notable loss in performance (i.e. urea yield rate and FE) if the tests were continued without these interruptions.

One explanation to this might be significant consumption of nitrate (CO_2 cannot be depleted as it is continuously supplied by purging the gas), although it is impossible to calculate this precisely since the

reviewer could not find the volume of the electrolyte solution used in the experiments (while it must be very prominently reported). Regardless, passing approximately 3.6 C of charge (1 h at ~1 mA) would consume at most (8e⁻; 100% FE) ~5 micromol of nitrate. This would make a detectable impact on the initial concentration of 0.1 M if only the electrolyte solution volume was less than 0.1 mL, which is highly unlikely.

Another explanation is poisoning of the catalyst with one of the products.

There might be also progressive changes to the catalyst resulting in the loss in the activity for the urea formation, i.e. substantial decrease in the FE. Significant reduction of Cu was indeed detected by XPS, although it was restored back to the original state upon oxidation with air oxygen. Hence, one might hypothesise that the catalyst cannot operate for a long period of time and requires periodic oxidation to restore its initial active state, as will likely happen during the change of the electrolyte solution (it is also noted that the authors do not describe how the electrolyte solution was changed and what was happening to the electrode during the change, although this must be clearly described in the paper).

However, by no means the reviewer and especially readers should be forced to make any of the above assumptions and hypothesise, but instead, the authors should:

- (i) report time-resolved yield rates and FEs for all key products during appropriately undertaken long-term experiments without any interruptions and contact of the electrode with air;
- (ii) if the above results in poor performance, the authors should explain that this can be resolved by periodic interruptions of the process and reoxidation of the catalyst (might be done electrochemically in situ?) to establish the initial state.
- (iii) discuss all of the above at the appropriate level of detail and in a highly transparent manner.
- (iv) undertake long-term experiments under optimal conditions (either with or without reoxidation) for a much longer period of time than very short 10 h tests to demonstrate accumulation of meaningful urea yields, i.e. at least an order of magnitude higher rather than ~1.5 micromol in each 1 h test (note that the solution was discarded after each step, i.e. synthesis of 15 micromol cannot be claimed).
- (v) demonstrate reproducibility of the longer-term test, i.e. repeat it 3 times with independent catalyst samples, not new electrodes modified with the same material.

2. Considering all the caveats of precise quantification of low concentrations of urea in electrolyte solutions (as explained in Ref.35, which is incorrectly cited as Ref.34 in line 134), it is important to confirm the amount of urea formed in the key long-term experiments under optimised conditions using direct NMR or HPLC/MS analysis of urea (not by analysing ammonia after urease treatment, which is also highly sensitive to the reaction conditions). Urea produces very distinct NMR signals, which can and should be quantified. Ideally, the authors should adopt a standard additions method when undertaking this experiment to ensure the lack of any reaction media effects.

3. Further on the urea analysis:

- (i) all UV-Vis data for the NH₃ analysis before and after urease treatment must be demonstrated on top of each other in the same plot and also as a difference spectra in a separate panel. Currently, these data are presented in a way that makes comparisons exceptionally difficult for a reader, which complicates assessment of the reliability of the results.
- (ii) Measurements of absorption $\gg 1$ (i.e. with the I/I₀ ratio way less than 10%) is highly unreliable and all of corresponding data should be remeasured by applying appropriate dilutions.

4. Error bars in figures are ascribed to "standard deviation of at least three independent measurements", but it is not appropriately explained what these measurements actually were. Were these repeats with the same electrode (as in Figure 3h)? Or repeats of tests with the same material, but freshly prepared electrodes? Or tests of three independent samples of the catalyst tested? All of these are important, and ideally all should be undertaken. First – to demonstrate stability of the electrode. Second – to demonstrate reproducibility of the electrolysis conditions. Third (the most important) – to demonstrate reproducibility of the electrolysis conditions and reproducibility of the catalytic properties. If the latter has not been done, the authors should undertake these tests under

optimised conditions and for the properly designed long-term tests.

5. Following from the comment above: the text must also use reliable performance metrics reported as mean +/- standard deviation with appropriate number of significant figures.

The yield rate of urea should be in the first place reported using (electro)chemical units, i.e. as mols per second per cm². The mass-normalised values can be still reported, but their significance is low given that the catalyst used is cheap, and nothing is known about its specific surface area.

6. "Recent studies demonstrate the feasibility of directly coupling CO₂ and N₂ reduction to form urea via heterogeneous electrocatalysis under ambient conditions⁶." – this statement is definitely false and must be removed. Direct electrochemical N₂ activation neither reductive (NH₃, CO(NH₂)₂) or oxidative (NO_x) has not been unambiguously proven in any paper published to date.

7. "easily obtained by non-thermal plasma activation" – this is misleading. Currently demonstrated energy-efficiencies of the cold-plasma N₂->NO process are very low, i.e. the process requires a lot of future developments to become practical. This does not mean that it will never become practical – we are just not there yet. This must be clearly explained.

8. "process is an ideal model" – nothing is ideal in a real experiment. Rerword.

9. "And Cu, W, and O were" – delete "And"

10. "highly ordered triclinic CuWO₄ and did not contain other impurities" – change to "detectable impurities"

11. Add more details on how LSVs were recorded, i.e. specify direction, how many scans were recorded, and which one is reported. Much more preferable and scientifically robust would be presentation of cyclic voltammograms, rather than archaic, highly ambiguous, semiquantitative and essentially useless linear sweeps.

It would be best if the authors have plotted the current densities after 1 h electrolysis at all potentials examined and showed these data in the main text (see <https://doi.org/10.1038/s41565-022-01121-4> if unsure), while LSVs could be in the SI.

12. NMR data should be plotted in the opposite direction of the ppm scale.

13. Figure S28 – what is the peak at ~5.7 ppm in the CO₂ data at the top?

14. Figure S29 – some detectable (?) urea formation is reported in the absence of CO₂, which most definitely needs to be explained. The reliable LOD for the employed method of analysis must be defined and reported.

15. Spectra in Fig.3c are very hard to read – each should be stretched vertically by at least a factor of 5. Why was it necessary to dilute the samples to the extent that the signal-to-noise ratio becomes so low?

16. The authors have done a perfect job on quantifying the mass-balance in their experiments, but strangely show this key important information in the SI. Figure S21 should be incorporated into Figure 3 (probably to replace the much less important panels 3f & 3g).

Plausible "loss" of ~10% of the FE should be explained.

17. What was the product distribution without CO₂ present? At least for the major CuWO₄ catalyst?

18. "simple reactions such as NO₃⁻ reduction to NO₂⁻ or NH₃" – nitrate to ammonia electroreduction is most definitely not a "simple" reaction.

19. Potentials should be discussed in terms of more/less positive/negative only, not "low/high" and similar.

20. "the CuO could effectively catalysis the reaction" – change catalysis to catalyse

21. As already mentioned above, the change in the Cu state after the experiment with CuWO₄ is significant, and it should be reported in the main text. Additionally, the authors should undertake similar experiment for the CuO control and discussed the similarities / differences.

22. The key features in Fig.4a are hardly seen – these spectra should be stretched vertically at least by a factor of 5 (carbon peak is irrelevant and can be off-scale). For example, the reviewer could hardly see the following change: "the peak intensities of *CO₃2– and *HCO₃– gradually decreased". It is also important to include the Ar data in this figure (currently in the SI). The "lack of space" cannot be an excuse, as reliable and clear demonstration of the data is always a top priority.

23. "As a non-time-resolved characterization method, the fast consumption of immediate(s) of no rate-determining steps hinders the observation of the corresponding signal; thus, the undissociated intermediates that served as substrates for the rate-determining step can usually be observed in the Raman spectra³⁷." – this is exceptionally hard to read and needs to be rephrased.

24. "The weak C≡O stretching peak" – is it really a triple bond?..

25. The interpretations and presentation of the TPD results are all problematic.

(i) "The NO₂-TPD curve of CuWO₄ also shows a series of NO₂ desorption peaks. However, the strength is weaker than that of WO₃," – what is the basis for this conclusion? The two curves exhibit the same signals, but with lower intensity for CuWO₄, which can be easily explained by the lower concentration of W.

(ii) "With the influence of octahedral [WO₆] clusters, the physical adsorption of CO by CuWO₄ significantly increased;" – the curves look very suspiciously identical to each other except for the offset. What was the background signal for these experiments? Was it reproducible?

(iii) "As shown in Supplementary Fig. 43b, CuO could be easily reduced, even below 100°C..." and "However, CuWO₄ has good low-temperature stability, indicating the Cu reaction sites could be stabilized by the [WO₆] clusters in CuWO" – all of these data appear highly random and in principle comparable to each other. The reviewer can hardly see how any robust conclusions can be drawn from these results.

26. Why the mass of 60 was not measured by DEMS? Urea can be most definitely detected by regular MS. This needs to be explained.

27. "identical to the results t that urea" – remove "t"

28. It is very interesting that NO and nitrite cannot be reduced into urea with the presented catalyst. Have the authors attempted probing the origins of this by DFT? That would be very important addition to the mechanistic, i.e. the most important and impactful, part of the manuscript.

29. It is hard to understand why the experimental is split between the main text and SI.

Most importantly, description of the characterisation techniques is unacceptably scant and must be substantially extended to allow readers to assess the reliability and reproduce the experiments. Sample preparation, handling, mounting procedures along with the key operational parameters and instrumental settings must be reported, especially for such highly sensitive techniques like XPS and in situ Raman. Some details on the electrochemical experiments are also missing, as mentioned above.

30. HUGE amount of data presented in the SI is currently very hard to navigate. The authors are

recommended to structure it in the format of separate sections each presenting only closely related data, e.g. 1. Physical characterisation, 2. Analytical methods, 3. In situ Raman spectroscopy, etc. A table of contents would be also highly useful.

Reviewer 1.

Reviewer's Comments:

In this manuscript, a CuWO_4 catalyst is used to synthesis urea from carbon dioxide and nitrate. The Faraday efficiency is 70.1% under a low operating potential (-0.2 V vs. RHE) and the urea production rate is high. The catalysts are well characterised and the possible reaction mechanism is explored. The manuscript should address the following question before it could be published.

Response:

First, we sincerely appreciate your positive comments on our work. Your constructive suggestions are very helpful for us to improve the quality of this work, and we have revised our manuscript accordingly.

Question 1:

From XRD results, there are other facets besides (111) facet. How these facets contribute to electrocatalysis processes?

Response:

Thank you for the comments. From XRD analysis, we observed that the (111) facet exhibited the strongest peak, suggesting its significant contribution to the reaction. Additionally, we investigated the mechanism on other facets by DFT as well. However, on the (100) facet (second highest peak), the reduction of $^*\text{COOH}$ to $^*\text{CO}$ led to surface reconstruction, which is contradicted by our experimental observations. the (110) facet was found unstable for prolonged existence compared to the (111) facet. Considering minimal contributions from other facets, despite the potential existence of alternative mechanisms, we chose to focus on reporting the most likely mechanism in this study.

Question 2:

The valence state of Cu is +2 in CuWO_4 while the lower valence state Cu involved during the electrosynthesis (page 10). How the lower valence state Cu contribute to electrocatalysis processes?

Response:

The Cu remains in a low-valence state during the reduction of CO_2 or NO_3^- is a commonly observed phenomenon (*Angew. Chem. Int. Ed.* 2021, 60, 17254-17267. *Angew. Chem. Int. Ed.* 2020, 59, 5350-5354. *J. Am. Chem. Soc.* 2018, 140, 8681-8689.). Most catalysts undergo reconfiguration after Cu valence changes just like the CuO reference catalyst in this work. However, we did not observe any significant structural changes in the CuWO_4 catalyst before and after electrocatalysis, as confirmed by various experimental techniques (XRD, TEM, In situ Raman). It indicates that the valence state

and structure of the Cu in the bulk phase have not changed. The observation of a lower valence state of Cu by Quasi-in-situ XPS (in Ar protection) could be attributed to the outermost layer of Cu^{2+} reduction reaction in electric potential, and it is likely to revert back to its original valence state after the electrolysis (Supplementary Fig. 39). Hence, the lower valence Cu is involved in the catalytic reaction in such a dynamical process. As the catalyst structure remained consistent throughout the reaction, no oxygen vacancies were deliberately introduced during catalyst preparation.

Question 3:

In Figure 5a, $^*\text{NO}_2 + ^*\text{CO}$ could form $^*\text{CONO}_2$, or $^*\text{HNO}_2 + ^*\text{CO}$. Here, $^*\text{CO}$ could also be reduced. What's the free energy if only $^*\text{CO}$ is reduced while $^*\text{NO}_2$ remains?

Response:

To address this concern, we have considered the further reduction of $^*\text{CO}$. As illustrated in Supplementary Fig. 52, the $^*\text{CO}$ can undergo hydrogenation to form $^*\text{CHO}$ or $^*\text{COH}$. However, the reaction free energies of $^*\text{CO}$ to $^*\text{CHO}$ (0.17 eV) and $^*\text{CO}$ to $^*\text{COH}$ (2.12 eV) are obviously higher compared to the C–N coupling process (−0.70 eV). Therefore, our proposed mechanism stands as reliable. Appropriate information and changes are added to the revised manuscript (page 18) and supporting documents (Supplementary Fig. 52).

Supplementary Figure 52. Free-energy diagram for CO hydrogenation and C–N coupling process.

Question 4:

When the first C–N bond formed, it's energy favoured. When the second C–N bond formed, it's energy unfavoured. Should the second C–N bond form with other intermediates, such as CONO₂, CONO, CONHO?

Response:

We appreciate the reviewer for raising this concern. To explore this possibility, we investigated various C–N coupling processes involving different intermediates, such as *CONO₂, *CONO, *CONHO, and *CONH (Supplementary Fig. 54). Among these, *CONO₂ was found to be unreactive towards *NO₂, preventing the formation of *CONO₂NO₂. Additionally, we compared these C–N coupling processes with their corresponding hydrogenation reactions, for example, *CONO + *NO₂ → *CONONO₂ compared to *CONO + H⁺ + e⁻ → *CONHO. It was evident that the formation of *CONONO₂, *CONHONO₂, and *CONHNO₂ was significantly more challenging compared to *CONHO, *CONHOH, and *CONH₂, respectively. These observations indicate that the energetics favor the formation of *CONHO, *CONHOH, and *CONH₂ over their corresponding C–N coupling products. Appropriate information and changes are added to the revised manuscript (page 19) and supporting documents (Supplementary Fig. 54).

Supplementary Figure 54. Free-energy diagram for the second C–N bond formed.

Question 5:

The free energy increases 0.93 eV from *NO₂ + *CO₂ to *NO₂ + *COOH, which is much larger than that of C–N bond formation between *NO₂ and *CO. Why the latter is the rate-determining step?

Response:

In electrochemical processes, both thermochemical and electrochemical elementary steps are involved. The step from $*NO_2 + *CO_2$ to $*NO_2 + *COOH$ involves proton-coupled electron transfer on the surface of electrode, which falls under the electrochemical step and can be potential-dependent. Hence, due to the highest reaction free energy, this step becomes the potential determining step in the process (J. Solid State Electrochem. 2013, 17, 339–344. Nat. Commun. 2021, 12, 4353. J. Phys. Chem. C 2017, 121, 26785–26793). On the other hand, the C–N bond formation step does not involve electron exchange with the electrode, making it a thermochemical step. Despite being exothermic, it still requires overcoming a 0.87 eV activation barrier. Therefore, it serves as the rate-determining step.

Question 6:

The lattice constants were calculated to be $a = 4.681$ angstrom, $b = 5.867$ angstrom, and $c = 4.898$ angstrom. Are these data from optimised structure?

Response:

The reported lattice constants in this work were obtained from the theoretically optimized structure. We have compared our data with several papers, including theoretical and experimental results, and found good agreement. Please refer to Table R1 for details.

Table R1. Comparison of the lattice constants.

Paper	Functional	a	b	c
This work	DFT+D3	4.681	5.867	4.898
	Hybrid DFT (HSE06)	4.665	5.840	4.896
Xie et al. ¹	DFT	4.780	6.000	4.930
Tian et al. ²	DFT+U (6 eV)	4.810	5.980	4.930
	experiment	4.690	5.830	4.870
Doumerc et al. ³	experiment	4.700	5.840	4.880
Forsyth et al. ⁴	experiment	4.690	5.830	4.880

1 Xie, X. et al. Efficient photo-degradation of dyes using $CuWO_4$ nanoparticles with electron sacrificial agents: a combination of experimental and theoretical exploration. RSC Advances 6, 953-959 (2016).

2 Tian, C. M. et al. Elucidating the electronic structure of $CuWO_4$ thin films for enhanced photoelectrochemical water splitting. J. Mater. Chem. A 7, 11895-11907 (2019).

3 Doumerc, J. P., Hejtmanek, J., Chaminade, J. P., Pouchard, M. & Krussanova, M. A photoelectrochemical study of $CuWO_4$ single crystals. Physica Status Solidi (a) 82, 285-294 (1984).

4 Forsyth, J. B., Wilkinson, C. & Zvyagin, A. I. The antiferromagnetic structure of copper tungstate, CuWO_4 . *J. Phys.: Condens. Matter* 3, 8433-8440 (1991).

Question 7:

It's hard to see Cu in Figure S46 and Figure S47.

Response:

We have included a zoomed-in picture in the Supporting Information that showcases both top and side views of the CuWO_4 (111) facet. The dimensions of the spheres representing the metals W and Cu are enlarged. The corresponding information was added to Supplementary Fig. 51.

Supplementary Figure 51. The construction of CuWO_4 (111) surfaces. Red, orange and cyan balls represent O, Cu, and W, respectively.

Reviewer 2.

Reviewer's Comments and Questions:

The authors prepare a CuWO_4 material which they use for simultaneous electroreduction of nitrate and CO_2 , suggesting that adsorbed intermediates recombine to form urea, still with low current density but relatively high faradaic efficiency. Even though I have concerns on how this system can lead to urea formation (comment 1 below), the authors do provide experimental evidence. Therefore, the manuscript could be published in

Nature Communications, but some comments should be addressed first:

Response:

We sincerely thank you for your time and effort on our manuscript, and we really appreciate that you agreed our work publishable after revision. Your constructive comments and questions are very helpful for us to revise and improve our manuscript. We have seriously studied all your comments, and tried our best to answer your questions and revise the manuscript accordingly.

Question 1:

The authors used an unbuffered electrolyte, so the alkaline interfacial environment that is created due to the H_2O , CO_2 and NO_3^- reduction reactions is likely turning CO_2 at the interface to carbonate, which is not reactive. Therefore, it is unclear how CO_2 is eventually present at the interface to be reduced to CO . In this electrolyte, I suspect that even the solution pH has increased. The weak buffer that is created using CO_2 gas, or the stirring of the electrolyte (as it looks from the image) will not be sufficient. I understand that the authors provide experimental data that support urea is formed, but on the contrary the hydroxide formation during the above reduction reactions and the CO_2 /bicarbonate/carbonate equilibria are unambiguous facts. The authors should very carefully consider the above and provide a clear explanation of their view of the interfacial conditions.

Response:

Thank you so much for your insightful comments. The pH stability of the reactive electrolyte and the transport form of CO_2 to the electrode surface are important for the CO_2 reduction process. Electrochemical coreduction of NO_3^- and CO_2 in H-cells with nitrate or HCO_3^- as electrolyte have been extensively reported in other works (*Nat. Sustain.* 2021, 4, 868-876. *ACS Nano* 2022, 16, 8213-8222. *Adv. Energy Mater.* 2022, 12, 2201500. *ACS Nano* 2022, 16, 9095-9104. *Appl. Catal. B: Environ.* 2022, 318, 121819.). We also concerned about this and monitored the pH of the electrolyte during electrolysis. We have performed controlled experiments using CO_3^{2-} and HCO_3^- as the carbon sources, which also produced urea (Fig. 4e), suggesting that carbonate can still be reactive for urea synthesis in our system. We also measured the pH of electrolyte in the reduction chamber before and after 20 hours of electrolysis. The pH of initial electrolyte was 5.3 due to the formation of carbonic acid from saturated CO_2 in the KNO_3 solution, and the carbonic acid/ HCO_3^- / CO_3^{2-} ionization equilibria could provide some buffering capacity. After 20 hours of electrolysis, the pH of the electrolyte in the reduction chamber was measured as 5.6, which is slightly raised (urea/ammonia water solution is slightly alkaline, or the mass transfer issue of Nafion film). The pH of the electrolyte can

be maintained because of that: 1) the anodic reaction (OER) could naturally consume the hydroxide formed during the cathodic reactions; 2) the continuous CO₂ flow and the ionization equilibriums provide a certain buffering capacity.

Question 2:

Elaborating a bit more on the choice of electrolyte, the authors state that the KNO₃ electrolyte was the “optimized electrolyte”. Please explain what you mean, what was it compared with? Note that since there is no supporting electrolyte, just KNO₃ which is reacting, its concentration is decreasing with time. A solid justification of the choice of electrolyte is needed, because it looks like the experiments were performed at conditions (nitrate concentration and pH) that were changing continuously.

Response:

We appreciate the reviewer’s curiosity about the original intention of the electrolyte concentration regulation experiment. For the co-reduction of nitrate and CO₂, the competitive adsorption between the two reactants occurs on the catalyst surface. In this work, we have balanced the adsorption capacity of the catalysts for both reactive species, which is crucial for the efficient synthesis of urea. Further regulation of the reactant concentration can investigate the adsorption equilibrium between the two reactants and find the optimal feeding ratio. The highest efficiency was obtained when the concentration ratio of the two reactive species is close to the ratio of carbon to nitrogen of urea itself, which is a sufficient proof that CuWO₄ can effectively balance the competition between the two species at the catalyst surface. The fluctuations in nitrate ion concentration and pH are negligible, due to the anodic reaction and the continuous formation of carbonic acid as a buffer. After a long period of electrolysis (20 h), a limited change (5.3 to 5.6) occurred in the electrolyte pH (Question 1). The change in nitrate concentration is even more insignificant, as only 3.6 C (1 mA cm⁻²) of electrons are transferred in an hour, even these electrons are 100% used for nitrate reduction (8 e⁻ to ammonia), consuming at most 5 micromoles of nitrate ions per hour. It is only about one thousandth of the amount of nitrate ions in the electrolyte (0.1 M, 35 mL).

Question 3:

The authors’ hypothesis is that they combine sites that reduce NO₃⁻ to NO₂ and other that reduce CO₂ to CO, and thereby they facilitate the NO₂+CO recombination, which they believe are the critical educts to form urea. I like the fact that the authors were driven by a hypothesis, but:

(a) The authors say if there were “continuous copper sites” this would lead to hydrocarbon formation. Are the authors sure that copper in their material is truly isolated? How can

they confirm they don't have adjacent copper atoms that will carry out the further reduction of CO?

(b) Copper is good for reducing nitrate to ammonia, but the whole analysis considers only tungsten responsible for nitrate reduction.

(c) It is unclear why the authors didn't use another material combination to support their arguments. For example, would it be possible to use silver instead of copper?

Response:

For the first one, the synthesized catalyst in our work is triclinic copper tungstate nanocrystals, which is confirmed by XRD and Raman characterization (Fig. 2a and Supplementary Fig. 1). No other crystal structure or impurities were detected, so the catalyst essentially maintained the standard copper-tungstate triclinic phase structure. In the copper-tungsten crystal structure, the copper and tungsten sites are arranged alternately (Fig. 2f), so we conclude that the copper sites are not continuous in general, especially in comparison to the copper oxide or copper catalyst.

For the second question, we are sorry for the description caused the reviewer's misunderstanding on the function of the bimetallic sites. Copper oxide has higher efficiency than tungsten oxide for nitrate reduction. But it can not stabilize reaction intermediates, easy to form nitrite and dissociate, which is not conducive to further C–N coupling. Tungsten oxide has the advantage of stabilizing the reaction intermediates of *NO_2 during nitrate reduction, but the *CO generation potential is high. Copper tungstate combines the advantages of both, which can reduce nitrate and stabilize the key intermediate *NO_2 . Throughout the catalytic process, copper tungstate works as a whole, rather than a single metal site only for one function.

For the third question, similar ideas were pursued at the beginning of our design. In addition to its excellent performance for CO₂ reduction, the more important reason for the final choice is that Cu is the most effective catalyst for C–C coupling by stabilizing intermediates such as *CO . And the *CO active intermediates also have the potential for C–N coupling. In contrast to copper, silver does not have the potential to further stabilize the key active intermediates in carbon-carbon or carbon-nitrogen coupling, and is more prone to CO formation and dissociation. We also prepared a silver tungstate catalyst and tested it. No nitrogen-carbon bond products were detected, confirming our initial conjecture.

Question 4:

How do the authors explain the higher current for CuO vs the CuWO₄ from the voltammetry? This contradicts with their interpretations.

Response:

Dear reviewer, first of all, current or current density is a parameter for the performance description of an electrode, rather than a catalyst, which involving the amount of catalyst, specific surface area, applied potential, activity, selectivity and so on. The higher current of CuO than that of CuWO₄ is not contradictory with the interpretations. The higher current does not mean better selectivity.

CuO has better catalytic properties for nitrate reduction and CO₂ reduction, especially for nitrate reduction, hence its higher current density. But, CuO is not able to stabilize reaction intermediates during nitrate reduction as shown in the in-situ Raman results (Supplementary Fig. 45), so CuO mainly converts nitrate to nitrite and ammonia instead of urea (Supplementary Fig. 23). Although the overall current density and nitrate reduction efficiency of WO₃ is low, it can stabilize the reaction intermediates during nitrate reduction (Supplementary Fig. 46), which facilitates further carbon-nitrogen coupling. Combining the properties of CuO and WO₃, CuWO₄ has a better ability to stabilize reactive intermediates than CuO and thus has a higher urea Faraday efficiency, although it sacrifices a certain current density.

Question 5:

DEMS is capable of detecting volatiles only. Therefore, the ammonia species detected in DEMS is different than those detected in the electrolyte with other methods. In addition, the authors' statement that NH₃ is detected in DEMS confirms that the solution is becoming alkaline, otherwise volatile ammonia would have not been formed in detectable amounts. Please consider both points and comment accordingly in the manuscript.

Response:

The electrochemical cell of DEMS has a highly permeable semi-permeable membrane and a vacuum negative pressure system. Dissolved gases and volatile organic compounds in the electrolyte can enter the mass spectrometer directly through the semi-permeable membrane. Between the working electrode and the semi-permeable membrane there is only a millimeter level of liquid film with finite dissolved CO₂. The local pH may rise due to the protons consumption and the generation of ammonia. For above reasons, ammonia was detected by DEMS. We hope our answer could dispel your concerns.

Question 6:

Why did the authors try to monitor CO at m/z 28? Given that the solution is saturated in CO₂, significant fragmentation will occur in the EI and will lead to a very high background for CO. Did the authors use 70 eV or softer ionization?

Response:

Thank you very much for your comment, and we are sorry for that we provided a wrong parameter of ionization in the initial manuscript. The differential electrochemical mass spectrometry (DEMS) test employed a high precision three-stage filter quadrupole mass analyzer mass analyzer with softer ionization (ionic energy: 4–150 eV). Hiden QMS has a unique soft ionization technology. By optimizing the tuning of the electron energy of the EI source, the fragmentation peak can be reduced and the prominent molecular ion peak can be strengthened, which achieves the purpose of reducing interference. And we regulated the potential with the switching cycles of open circuit and working states, which presents a more efficient contrast between the test and background signals. Therefore, the tested CO signal at m/z 28 has a higher reliability. We have corrected the corresponding information in the revised manuscript (page 24).

Question 7:

A last comment on the concept description in first paragraphs of the intro. I think it gives an incomplete picture; the urea market is globally very large and the process described by the authors would require enormous amounts of feedstocks, i.e. CO₂ and nitrate. Regarding CO₂, the intro now neglects that industrial urea synthesis is integrated with ammonia synthesis with grey hydrogen, so there is a point source of CO₂. Regarding nitrate, (which is currently made by ammonia) the authors state that it can originate from wastewater (but concentrations are very low) or plasma oxidation of N₂ (which is however energy intensive). A bit more balanced discussion here on the scale challenges would be more accurate.

Response:

We fully agree with the reviewer's viewpoint of large-scale production of urea with nitrate as a feedstock. We have revised this part of the manuscript to make it more accurate (page 2).

Reviewer 3.**Reviewer's Comments:**

The manuscript by Zhao et al. presents a clear hypothesis on the design of an electrocatalysts for the conversion of nitrate and carbon dioxide to urea (Figure 1 is great) and confirm it with an impressive set of experimental and computational data. The mechanistic studies presented by the authors might be of particularly high value to the

field.

In principle, the manuscript might be suitable for publication in Nature Communications, if the authors can address the key comments on the stability of the CuWO_4 catalyst and urea analysis. Otherwise, the paper should be published in a more specialised journal upon addressing the comments below.

Response:

First of all, we sincerely appreciate your time and effort for our work. You have given many constructive suggestions and comments, even on the details and language, which are greatly helpful. We have seriously studied all your comments, and performed some additional experiments accordingly to address your concerns. We sincerely hope that our answers can address your concerns, and please consider our manuscript for publication.

Question 1

Why was it necessary to change the solution after every 1 h of electrolysis in Fig. 3h? The only reason the reviewer could think of is a notable loss in performance (i.e. urea yield rate and FE) if the tests were continued without these interruptions.

One explanation to this might be significant consumption of nitrate (CO_2 cannot be depleted as it is continuously supplied by purging the gas), although it is impossible to calculate this precisely since the reviewer could not find the volume of the electrolyte solution used in the experiments (while it must be very prominently reported). Regardless, passing approximately 3.6 C of charge (1 h at ~ 1 mA) would consume at most ($8e^-$; 100% FE) ~ 5 micromol of nitrate. This would make a detectable impact on the initial concentration of 0.1 M if only the electrolyte solution volume was less than 0.1 mL, which is highly unlikely.

Another explanation is poisoning of the catalyst with one of the products.

There might be also progressive changes to the catalyst resulting in the loss in the activity for the urea formation, i.e. substantial decrease in the FE. Significant reduction of Cu was indeed detected by XPS, although it was restored back to the original state upon oxidation with air oxygen. Hence, one might hypothesise that the catalyst cannot operate for a long period of time and requires periodic oxidation to restore its initial active state, as will likely happen during the change of the electrolyte solution (it is also noted that the authors do not describe how the electrolyte solution was changed and what was happening to the electrode during the change, although this must be clearly described in the paper).

However, by no means the reviewer and especially readers should be forced to make any of the above assumptions and hypothesise, but instead, the authors should:

(i) report time-resolved yield rates and FEs for all key products during appropriately undertaken long-term experiments without any interruptions and contact of the electrode

with air;

(ii) if the above results in poor performance, the authors should explain that this can be resolved by periodic interruptions of the process and reoxidation of the catalyst (might be done electrochemically in situ?) to establish the initial state.

(iii) discuss all of the above at the appropriate level of detail and in a highly transparent manner.

(iv) undertake long-term experiments under optimal conditions (either with or without reoxidation) for a much longer period of time than very short 10 h tests to demonstrate accumulation of meaningful urea yields, i.e. at least an order of magnitude higher rather than ~1.5 micromol in each 1 h test (note that the solution was discarded after each step, i.e. synthesis of 15 micromol cannot be claimed).

(v) demonstrate reproducibility of the longer-term test, i.e. repeat it 3 times with independent catalyst samples, not new electrodes modified with the same material.

Response:

Thank you very much for your comments. Catalyst stability is a very important issue. In the previous experiment, we used a conventional H-type electrolytic cell for testing, which lacks a sampling port (Supplementary Fig. 10). In order to monitor the FE on an hourly basis, we need to interrupt the test and remove a portion of the electrolyte for testing, and new electrolyte was added each time. But such testing patterns may create confusion for reviewers and readers. Therefore, we purchased a new H-type electrolytic cell with a sampling port (Supplementary Fig. 10) and performed an uninterrupted 20-hour electrolytic test. To prevent a obvious drop of liquid level, the electrolyte was extracted only at 10 h and 20 h for quantitative tests of urea. In order to demonstrate the repeatability of the longer-term test, we repeated the 20 h electrolysis test three times with the same electrode (Supplementary Fig. 35), as suggested by the reviewer. Based on the data of longer-term stability, the catalyst did not lose the activity in the continuous and uninterrupted electrolysis, and the final Faraday efficiency remained above 50% after three longer-term repeated tests. The total FE for the first ten hours of continuous electrolysis is 64%, which is close to the average Faraday efficiency (68%) of the previous 10 h electrolysis that changed the electrolyte every hour. It indicates that the catalyst does not require an oxidation regeneration. There is a 15%–20% loss of current density after the three longer-term electrolysis, which may be caused by catalyst shedding, partial inactivation or other environmental factors. In all tests, the urea was quantified by ¹H-NMR (Supplementary Fig. 34 and 35). Although the continuous electrolysis we provided is relatively limited in time dimension, it generally reflects the stability of our catalyst. In order to obtain longer lasting catalytic stability, the method of electrode preparation and catalyst loading need to be improved. We will make further efforts in future works. The

results have been added to Supplementary Fig. 10, 34 and 35, and related discussions have been added to the revised manuscript on page 10.

Supplementary Figure 10 The digital images of the H-cell and H-cell with sampling ports for electrochemical measurements.

Supplementary Figure 34 (a) ¹H NMR spectra of ¹⁴N-urea with various concentrations. (b) The calibration curves for ¹⁴N-urea.

Supplementary Figure 35. (a, c, e) three long-term repeated tests of urea synthesis during 20 h of electrolysis at -0.2 V vs. RHE in 0.1 M KNO_3 with CO_2 bubbling (20 mL min^{-1}) and (b, d, f) ^1H NMR data of corresponding electrolyte.

Question 2

Considering all the caveats of precise quantification of low concentrations of urea in electrolyte solutions (as explained in Ref. 35, which is incorrectly cited as Ref. 34 in line 134), it is important to confirm the amount of urea formed in the key long-term experiments under optimised conditions using direct NMR or HPLC/MS analysis of urea (not by analysing ammonia after urease treatment, which is also highly sensitive to the reaction conditions). Urea produces very distinct NMR signals, which can and should be quantified. Ideally, the authors should adopt a standard additions method when undertaking this experiment to ensure the lack of any reaction media effects.

Response:

The quantitative method of urea is the most important key to the reliability of related research. Ref. 35 has a systematic discussion of the quantitative approach to urea and the corresponding interference factors. The interference of nitrite concentration on the

determination of urea yield by the DAMO-TSC method is also discussed in detail in Ref. 34. The relevant information is placed in the supporting information, so it is not easy to find. Therefore, it is more appropriate to cite Refs. 34 and 35 together. For the urea quantification in this work, we employed the NMR quantification method for the urea yield of the catalyst, which is consistent with the reviewer's expectation. In addition, the ^1H -NMR quantification of urea after ^{15}N isotope labeling was performed in this work (Fig. 3c and Supplementary Fig. 28), avoiding the effect of other nitrogenous pollution (*Chem. Eng. J.* 2023, 453, 139836; *Small Methods* 2022, 6, 2200561.). The quantification of urea in supplementary longer-term stability tests also used NMR methods to ensure the reliability of the results, which have been shown in the response to Question 1. The results have been added to Supplementary Fig. 34 and 35, and related discussions were made in the revised manuscript on page 10.

Question 3

Further on the urea analysis:

- (i) all UV-Vis data for the NH_3 analysis before and after urease treatment must be demonstrated on top of each other in the same plot and also as a difference spectra in a separate panel. Currently, these data are presented in a way that makes comparisons exceptionally difficult for a reader, which complicates assessment of the reliability of the results.
- (ii) Measurements of absorption $\gg 1$ (i.e. with the I/I_0 ratio way less than 10%) is highly unreliable and all of corresponding data should be remeasured by applying appropriate dilutions.

Response:

First of all, thank you for the suggestion. We have placed the UV-visible spectra before and after the urease treatment in the same figure for the reader's more intuitive comparison. And we agree with the your suggestion that measurements of absorption $\gg 1$ is not reliable. All UV-Vis data for the NH_3 analysis have been were remeasured by applying appropriate dilutions. Revised Supplementary Fig. 18, 20, 22, 24 and 31, etc. have been added to the revised manuscript and supporting information.

Supplementary Figure 18 Absolute calibration of the indophenol blue method for quantification of NH_3 estimated by NH_4^+ ion concentration. (a) UV-Vis spectra of NH_3 with various concentrations. (b) The calibration curve for NH_3 .

Supplementary Figure 20 Chromatographic curves (a, b) and UV-vis absorption spectra (c) of urea quantified by urease decomposition method for CuWO_4 based on ion chromatography and the indophenol blue method.

Supplementary Figure 22 Chromatographic curves (a, b) and UV-vis absorption spectra (c) of urea quantified by urease decomposition method for CuO based on ion chromatography and the indophenol blue method.

Supplementary Figure 24 Chromatographic curves (a, b) and UV-vis absorption spectra (c) of urea quantified by urease decomposition method for WO_3 based on ion chromatography and the indophenol blue method.

Supplementary Figure 31 CuWO_4 electrolysis in 0.1 M KNO_3 electrolyte with Ar bubbling at the corresponding potential for 1 hour. (a) Chrono-amperometry results of CuWO_4 in 0.1 M KNO_3 with Ar bubbling at the corresponding potentials. (b) UV-Vis spectra of the electrolyte before and after urease decomposition with indophenol indicator. (c) NH_3 yield rate at different potentials for CuWO_4 . (d) Urea yield rate on CuWO_4 at different applied potentials. (e) Faraday efficiency of different products for CuWO_4 at different applied potentials.

Question 4

Error bars in figures are ascribed to “standard deviation of at least three independent measurements”, but it is not appropriately explained what these measurements actually were. Were these repeats with the same electrode (as in Figure 3h)? Or repeats of tests with the same material, but freshly prepared electrodes? Or tests of three independent samples of the catalyst tested? All of these are important, and ideally all should be undertaken. First- to demonstrate stability of the electrode. Second- to demonstrate reproducibility of the electrolysis conditions. Third (the most important) to demonstrate reproducibility of the electrolysis conditions and reproducibility of the catalytic properties. If the latter has not been done, the authors should undertake these tests under optimised conditions and for the properly designed long-term tests.

Response:

Thanks for the reviewer's suggestions. The error-bar data in this paper were obtained from three freshly prepared electrodes with the same material. This mainly demonstrated the reproducibility of electrolysis conditions and the reproducibility of catalytic performance. In Fig. 3h, we conducted ten repeatability tests on the same electrode, indicating that the electrode material has excellent stability and repeatability of catalytic properties. In addition, as suggested by the reviewers, we added 20 h continuous electrolysis and repeated three times to explore the longer-term stability and repeatability of the catalytic performance (Supplementary Fig. 34 and 35).

Question 5

Following from the comment above: the text must also use reliable performance metrics reported as mean +/- standard deviation with appropriate number of significant figures. The yield rate of urea should be in the first place reported using (electro)chemical units, i.e. as mols per second per cm². The mass-normalised values can be still reported, but their significance is low given that the catalyst used is cheap, and nothing is known about its specific surface area.

Response:

The catalysts prepared in this work are powder catalysts. In electrode preparation, we control the area of the electrode coating (1 cm⁻²) and the mass of catalyst (1 mg). Calculated according to the electrode area, the unit of yield can be directly converted to ug h⁻¹ cm⁻²_{electrode area} without numerical change. However, it is also a rough expression form. In order to better compare the samples, the electrochemical active surface area (ECSA) measurements were carried out. The ECSA of CuWO₄ electrode is slightly higher than that of CuO and WO₃. After ECSA normalization, the CuWO₄ electrode still has the highest urea yield compared to the other two. The results have been added to

Supplementary Fig. 26 and 27, and related discussions are made in the revised manuscript on page 7.

Supplementary Figure 26 Determination of the ECSA for the CuWO_4/CFP , CuO/CFP and WO_3/CFP electrodes. CV curves of (a) CuWO_4/CFP , (c) CuO/CFP and (e) WO_3/CFP in 0.1 M KNO_3 with different scan rates at selected potential range. The corresponding capacitance $\Delta j (|j_{\text{charge}} - j_{\text{discharge}}|)$ of (b) CuWO_4/CFP , (d) CuO/CFP and (f) WO_3/CFP electrodes versus the scan rates. The scanning potential range is from 0.3 V to 0.42 V vs RHE. ECSA of electrode was obtained from CV curves, in details, by plotting the $\Delta j (|j_{\text{charge}} - j_{\text{discharge}}|)$ at Faradaic silence potential range against the scan rates, the linear slope is obtained, which is a positive correlation with the double-layer capacitance (Cdl), and been used to represent the corresponding ECSA.

Supplementary Figure 27 (a) The LSV curves and (b) urea yield normalized by ECSA of CuWO₄/CFP, CuO/CFP and WO₃/CFP electrodes.

Question 6

“Recent studies demonstrate the feasibility of directly coupling CO₂ and N₂ reduction to form urea via heterogeneous electrocatalysis under ambient conditions⁶.” this statement is definitely false and must be removed. Direct electrochemical N₂ activation neither reductive (NH₃, CO(NH₂)₂) or oxidative (NO_x) has not been unambiguously proven in any paper published to date.

Response:

After carefully investigating published literatures, we agree with your opinion that “Direct electrochemical N₂ activation neither reductive (NH₃, CO(NH₂)₂) or oxidative (NO_x) has not been unambiguously proven in any paper published to date.” We have revised the statement in the revised manuscript (page 2).

Question 7

“easily obtained by non-thermal plasma activation” this is misleading. Currently demonstrated energy-efficiencies of the cold-plasma N₂->NO process are very low, i.e. the process requires a lot of future developments to become practical. This does not mean that it will never become practical, we are just not there yet. This must be clearly explained.

Response: Thanks to the reviewer's suggestion, We have revised the related description on page 2.

Question 8

“process is an ideal model” nothing is ideal in a real experiment. Reword.

Response: We have revised the related description on page 2.

Question 9

“And Cu, W, and O were” delete “And”

Response:

Thank you very much for your careful reading, and such errors have been corrected in the revised manuscript.

Question 10

“highly ordered triclinic CuWO_4 and did not contain other impurities” change to “detectable impurities”

Response:

Thanks for the suggestion, it makes the statement more accurate. We have revised the manuscript accordingly.

Question 11

Add more details on how LSVs were recorded, i.e. specify direction, how many scans were recorded, and which one is reported. Much more preferable and scientifically robust would be presentation of cyclic voltammograms, rather than archaic, highly ambiguous, semiquantitative and essentially useless linear sweeps. It would be best if the authors have plotted the current densities after 1 h electrolysis at all potentials examined and showed these data in the main text (see <https://doi.org/10.1038/s41565-022-01121-4> if unsure), while LSVs could be in the SI.

Response:

The LSV measurements were performed with a negative scan direction and a scan speed of 10 mV/s. The final stable curves were recorded after multiple scans (10–15 times). The dominant potential for urea formation can be tentatively identified by comparing LSV curves with argon and CO_2 . Therefore, the LSV curves not only represent the relative dependence of the current density between different samples, but is also an important data for determining the initial dominant potential interval. The current densities for different potentials should also be shown more precisely in the main text, so we have added the *I*-*V* plots in Fig. 3a as suggested by the reviewer.

Fig. 3a LSV curves of CuWO₄, CuO, and WO₃ in 0.1 M KNO₃ with Ar or CO₂ bubbling and *I*-*V* plots of CuWO₄, CuO, and WO₃ in 0.1 M KNO₃ with CO₂ at different potentials.

Question 12

NMR data should be plotted in the opposite direction of the ppm scale.

Response:

Thanks to the reviewer for the comments. We have made corrections accordingly. The Fig. 3c and Supplementary Fig. 26–28 have been corrected.

Question 13

Figure S28 what is the peak at ~5.7 ppm in the CO₂ data at the top?

Response:

The peak at 5.7 ppm in the ¹H NMR is attributed to ¹⁴N-urea. The ¹H NMR signal of ¹⁵N-urea is a double peaks between 5.6 ppm and 5.8 ppm. (*Nat. Chem.* 2020, 12, 717-724.)

Question 14

Figure S29 some detectable urea formation is reported in the absence of CO₂, which most definitely needs to be explained. The reliable LOD for the employed method of analysis must be defined and reported.

Response:

The results in the argon condition were calculated by detecting the concentration of ammonium ions by the indophenol blue method after urease decomposition. Although we have tried our best to control variables, the introduction of new reagents during urease degradation of urea may still affect the final colorimetric result of ammonium ion. When the overall concentration of ammonium ion increases with the potential, the fluctuation

error was also magnified. So we also diluted the electrolyte for UV-vis detection according to the reviewer's suggestion in Question 3, and the error was significantly reduced (less than 1 ug) (Supplementary Fig. 31). In addition, it is more reasonable to judge the urea content by NMR as compared to urea decomposition and other colorimetric methods. In this work, the main yields of urea were quantitatively analyzed by NMR.

Supplementary Figure 31 CuWO₄ electrolysis in 0.1 M KNO₃ electrolyte with Ar bubbling at the corresponding potential for 1 hour. (a) Chrono-amperometry results of CuWO₄ in 0.1 M KNO₃ with Ar bubbling at the corresponding potentials. (b) UV-Vis spectra of the electrolyte before and after urease decomposition with indophenol indicator. (c) NH₃ yield rate at different potentials for CuWO₄. (d) Urea yield rate on CuWO₄ at different applied potentials. (e) Faraday efficiency of different products for CuWO₄ at different applied potentials.

Question 15

Spectra in Fig.3c are very hard to read – each should be stretched vertically by at least a factor of 5. Why was it necessary to dilute the samples to the extent that the signal-to-noise ratio becomes so low?

Response:

The electrolyte was not diluted. It is the original concentration after tests. To make it easier to read, we have enlarged it by a factor of 2 in the vertical direction, which is clear enough. In order to ensure that the scale is consistent, we also made the same adjustment to Supplementary Figure 28a.

Fig. 3c. ^1H NMR data of isotope calibration experiment in 0.1 M K^{15}NO_3 with CO_2 bubbling (20 mL min^{-1}) at different applied potentials.

Supplementary Figure 28 NMR spectra and calibration curves for quantification of ^{15}N -urea and $^{15}\text{NH}_4^+$. (a) NMR spectra of ^{15}N -urea with various concentrations. (b) The calibration curves for ^{15}N -urea. (c) NMR spectra of $^{15}\text{NH}_4^+$ with various concentrations. (d) The calibration curves for $^{15}\text{NH}_4^+$.

Question 16

The authors have done a perfect job on quantifying the mass-balance in their experiments, but strangely show this key important information in the SI. Figure S21 should be incorporated into Figure 3 (probably to replace the much less important panels 3f & 3g). Plausible “loss” of ~10% of the FE should be explained.

Response:

We accept the reviewer's recommendation after careful consideration. Fig. 3g is sufficient to provide the main information on the regulation of the concentrations of the two reactants. So we have moved Fig. 3f to Supplementary Fig. 32 and added the more important Figure S21c to Fig. 3e. Thanks for the reviewer's suggestion, which may be more convenient for readers to get the main information.

The loss of nearly 10% of Faraday's efficiency can be attributed to several reasons. Such as, a small amount of gaseous product dissolves in the electrolyte or escapes during collection; or some of the liquid phase products may be adsorbed on the electrode surface or permeated through the Nafion membrane.

Fig. 3 | Electrochemical synthesis of urea. **a** LSV curves of CuWO₄, CuO, and WO₃ in 0.1 M KNO₃ with Ar or CO₂ bubbling and I-V plots of CuWO₄, CuO, and WO₃ in 0.1 M KNO₃ with CO₂ at different potentials. **b** Yield rates and FE values of urea production for CuWO₄ at different applied potentials in 0.1 M KNO₃ with CO₂ bubbling (20 mL min⁻¹). **c** ¹H NMR data of isotope calibration experiment in 0.1 M K¹⁵NO₃ with CO₂ bubbling (20 mL min⁻¹) at different applied potentials. **d** ¹⁵N-urea yield rates and FEs via integrated peak area from NMR data. **e** FE values of all products for CuWO₄ at different applied potentials in 0.1 M KNO₃ with CO₂ bubbling (20 mL min⁻¹). **f** N_{urea}-selectivity and NO₃⁻RR-FE for CuWO₄ at different applied potentials in 0.1 M KNO₃ with CO₂ bubbling (20 mL min⁻¹). **g** Urea FEs for CuWO₄ at -0.2 V vs. RHE in 0.1 M KNO₃ in different concentrations of KNO₃ electrolyte with CO₂ bubbling (20 mL min⁻¹) (**g**). **h** Stability test of urea synthesis during 10 h of electrolysis at -0.2 V vs. RHE in 0.1 M KNO₃ with CO₂ bubbling (20 mL

min⁻¹). **i** Comparison of the results of this work with state-of-art electrocatalytic synthesis urea catalysts in terms of operation potential and FE. **b,d–g** Error bars in accordance with the standard deviation of at least three independent measurements.

Question 17

What was the product distribution without CO₂ present? At least for the major CuWO₄ catalyst?

Response:

We are sorry that we omitted this important information in our previous manuscript. At the beginning of the study, we carefully investigated the product distribution in the absence of CO₂. The products are mainly nitrite, ammonia, and hydrogen. We have added the corresponding results to Supplementary Fig. 31.

Supplementary Figure 31 (e) Faraday efficiency of different products for CuWO₄ in 0.1 M KNO₃ electrolyte with Ar bubbling at different applied potentials.

Question 18

“simple reactions such as NO₃⁻ reduction to NO₂⁻ or NH₃” nitrate to ammonia electroreduction is most definitely not a “simple” reaction.

Response:

Thanks to the reviewer's comments, we have changed “simple reactions” to “side reactions” (page 9).

Question 19

Potentials should be discussed in terms of more/less positive/negative only, not “low/high” and similar.

Response:

It is a good suggestion for the accuracy of the language used in the article. We have checked and revised the manuscript throughout.

Question 20

“the CuO could effectively catalysis the reaction” change catalysis to catalyse

Response:

Thank you very much for your careful reading and this error has been corrected in the revised manuscript. We have checked the manuscript carefully and fixed other errors (page 9).

Question 21

As already mentioned above, the change in the Cu state after the experiment with CuWO_4 is significant, and it should be reported in the main text. Additionally, the authors should undertake similar experiment for the CuO control and discussed the similarities / differences.

Response:

As the reviewer mentioned, the information on the variation of the Cu valence is important, which is often ignored. We have added the change in Cu valence state of CuO catalyst before and after the reaction. After electrolysis, copper also changed from a Cu^{2+} to a lower valence state, indicating that Cu^+ catalyzed the reaction on CuO. It is consistent with the results of CuWO_4 . However, the valence state of Cu can not change completely back to Cu^{2+} after exposure to air, probably because the structure of CuO has been reconstructed, which in a line with the result of in-situ Raman spectra of CuO (Supplementary Figure 42). We can not put these XPS data in the main text due to the space limitation, we hope for your understanding. The XPS results have been added to Supplementary Fig. 43, and related discussions have been added in the revised manuscript on pages 10 and 11.

Supplementary Figure 43 Cu 2p XPS spectra of CuO before and after catalysis reaction (in Ar protection or in the air).

Question 22

The key features in Fig. 4a are hardly seen – these spectra should be stretched vertically at least by a factor of 5 (carbon peak is irrelevant and can be off-scale). For example, the reviewer could hardly see the following change: “the peak intensities of CO_3^{2-} and HCO_3^- gradually decreased”. It is also important to include the Ar data in this figure (currently in the SI). The “lack of space” cannot be an excuse, as reliable and clear demonstration of the data is always a top priority.

Response:

Thanks to the reviewer's suggestion, we have enlarged Fig. 4a and added the Ar data.

Fig. 4 a) In-situ Raman spectra of CuWO_4 in 0.1 M KNO_3 with CO_2 bubbling at different applied potentials or Ar bubbling at open circuit state.

Question 23

“As a non-time-resolved characterization method, the fast consumption of immediate(s) of no rate-determining steps hinders the observation of the corresponding signal; thus, the undissociated intermediates that served as substrates for the rate-determining step can usually be observed in the Raman spectra 37.” this is exceptionally hard to read and needs to be rephrased.

Response:

Thanks for the reviewer's comments. In order to facilitate readers' understanding, we have revised the statement as follows: “As reaction intermediates irrelevant to the rate-determining step, and the product of rate-determining step could be rapidly consumed, which are difficult to be observed by non-time-resolved characterization methods. Thus, the undissociated intermediates that serve as substrates for the rate-determining step can be observed in Raman spectra.” (page 13)

Question 24

“The weak $\text{C}\equiv\text{O}$ stretching peak” is it really a triple bond?.

Response:

It is an important detail. Although it is labeled as $\text{C}\equiv\text{O}$ bond in a few literatures (*J. Phys. Chem. C* 2019, 123, 5951-5963), the bond structure of *CO intermediates from CO_2 reduction should be different from the $\text{C}\equiv\text{O}$ bond of carbon monoxide. So it is not accurate to describe it in terms of $\text{C}\equiv\text{O}$. Referring to the literatures (*Nat. Commun.* 2022, 13, 2656; *Nat. Commun.* 2021, 12, 3264; *Anal. Chem.* 2022, 94, 11337-11344), the vibration peak

of adsorbed *CO species is more accurate. The corresponding part of the manuscript has been revised (page 13).

Question 25

The interpretations and presentation of the TPD results are all problematic.

(i) “The NO₂-TPD curve of CuWO₄ also shows a series of NO₂ desorption peaks. However, the strength is weaker than that of WO₃,” – what is the basis for this conclusion? The two curves exhibit the same signals, but with lower intensity for CuWO₄, which can be easily explained by the lower concentration of W.

(ii) “With the influence of octahedral [WO₆] clusters, the physical adsorption of CO by CuWO₄ significantly increased;” – the curves look very suspiciously identical to each other except for the offset. What was the background signal for these experiments? Was it reproducible?

(iii) “As shown in Supplementary Fig. 43b, CuO could be easily reduced, even below 100° C...” and “However, CuWO₄ has good low-temperature stability, indicating the Cu reaction sites could be stabilized by the [WO₆] clusters in CuWO” – all of these data appear highly random and in principle comparable to each other. The reviewer can hardly see how any robust conclusions can be drawn from these results.

Response:

Thank you for the comments. Our responses to the comments are described below in a point-to-point manner.

- (i) NO₂-TPD uses mass spectrometry to analyze the desorbed gas. All samples were tested under the same conditions, including adsorption gas concentration, adsorption time, purge duration, background gas, etc. The desorption peak is in the same coordinate scale for all three samples, and the peak intensity represents the amount of desorbed NO₂ gas at corresponding temperature. It can be seen from Figure 4b that the desorption peak intensity of CuWO₄ sample is weaker than that of WO₃. Note that we described changes in the strength of the desorption peak, not the strength of the adsorption capacity. The level of desorption peak indicates that the amount of NO₂ adsorbed on CuWO₄ surface is lower than that of WO₃. Naturally, this may be caused by the content of W or the strength of adsorption.
- (ii) Consistent with the previous question, all samples were tested under consistent conditions, so that the desorption peak can represent the amount of desorbed gas. All of the tests were conducted using high purity argon gas as the background and repeatable. In Supplementary Fig. 47a, the desorption peak below 150°C represents the amount of physically adsorbed CO. The height of the desorption peak of CuWO₄ is significantly higher than CuO, indicating that more CO is

absorbed on the CuWO₄ by physical adsorption.

- (iii) The generation of CO₂ mainly comes from the reduction reaction of CO to metal oxides. We would like to illustrate the difficulty of reducing metal oxides by comparing the amount of CO₂ produced with the temperature. However, this cannot be used directly as the evidence for the stability of CuWO₄. Hence, we have removed and modified this part. Thanks again to the reviewer for the comments.

Question 26

Why the mass of 60 was not measured by DEMS? Urea can be most definitely detected by regular MS. This needs to be explained.

Response:

DEMS mainly test dissolved gas molecules and volatile organic compounds using negative pressure chambers and gas semi-permeable membranes. Urea is soluble in water and not volatile. Thus the DEMS did not detect significant signal fluctuations at mass of 60. To confirm the urea formation in the DEMS test, a qualitative test for urea was performed on the post-test electrolyte, and the result is positive.

Question 27

“identical to the results t that urea” remove “t”

Response:

Thank you very much for your careful reading and this error has been corrected in the revised manuscript.

Question 28

It is very interesting that NO and nitrite cannot be reduced into urea with the presented catalyst. Have the authors attempted probing the origins of this by DFT? That would be very important addition to the mechanistic, i.e. the most important and impactful, part of the manuscript.

Response:

We have conducted a series of experiments to validate this conclusion. Additionally, to further verify our observation, we performed the initial hydrogenation step of *NO₂. By comparing the reaction free energy of *NO₂ + H⁺ + e⁻ → *HNO₂ and *NO₂ + *CO → *CONO₂, we observed that the C–N coupling of *NO₂ and *CO is more exothermic than the hydrogenation of *NO₂. This suggests that the C–N coupling reaction of *NO₂ and *CO is easier and faster. These findings are strongly in agreement with our experimental observations.

Question 29

It is hard to understand why the experimental is split between the main text and SI. Most importantly, description of the characterisation techniques is unacceptably scant and must be substantially extended to allow readers to assess the reliability and reproduce the experiments. Sample preparation, handling, mounting procedures along with the key operational parameters and instrumental settings must be reported, especially for such highly sensitive techniques like XPS and in situ Raman. Some details on the electrochemical experiments are also missing, as mentioned above.

Response:

Thank you for your comments, we would like to provide more detailed experimental information. Details of the sample preparation, the electrochemical measurements, quantitative analysis method, in situ characterization and operating parameters of each instruments have been added to the Method section of the maintext. To shorten the length of the manuscript, the detailed list of chemicals and characterization equipments were placed in the Supporting Information.

Question 30

HUGE amount of data presented in the SI is currently very hard to navigate. The authors are recommended to structure it in the format of separate sections each presenting only closely related data, e.g. 1. Physical characterisation, 2. Analytical methods, 3. In situ Raman spectroscopy, etc. A table of contents would be also highly useful.

Response:

This is a particularly good suggestion. We have classified the data in the Supporting Information according to their functions, and provided a more detailed table of contents.

REVIEWERS' COMMENTS

Reviewer #2 (Remarks to the Author):

I believe that the revised version is suitable for publication.

Reviewer #3 (Remarks to the Author):

Zhao et al. have invested appropriate efforts into addressing the reviewers' comments from the first round, which produced a substantially improved manuscript, which can be recommended for publication subject to correction of the remaining minor, generally technical, issues. No further review is necessary from my side.

- (1) "Faraday efficiency" -> "faradaic efficiency" (see Bard's electrochemistry dictionary, if unsure).
- (2) The performance metrics should be reported in a statistically meaningful manner throughout the manuscript, and especially in the abstract and conclusions, i.e. as mean +/- standard deviation with appropriate number of significant figures.
- (3) Lines 47-48: "Moreover, the lower dissociation energy of the nitrogen-oxygen bond (204 kJ mol⁻¹) eases the coupling of NO₃⁻ reduction with CO₂ reduction to accomplish urea electrosynthesis from a thermodynamic perspective" – I would recommend removing the "thermodynamic perspective", as this is more about activation energy, i.e. kinetics.
- (4) Line 76: "reduction at a relative positive potential" – change to e.g. "positive potentials", or better "practical overpotentials" or "low overpotentials".
- (5) Line 133: "CuWO₄ and CuO increased significantly at the potential range from -0.1 to -0.4 V vs. RHE" – it is a bit of a stretch to refer to the observed increase as significant. Better delete this word
- (6) Figure captions should be much more detailed, and at minimum report the composition of the electrolyte solution and the gas used. For example, it is impossible to understand which experiments are shown in Figure S11, and quite a few other figures as well.
- (7) Lines 138-140: "and the production of urea was quantified by diacetylmonoxime-thiosemicarbazide (DAMO-TSC) and nuclear magnetic resonance (NMR) methods (Supplementary Fig. 12)." – Figure S12 shows a calibration for the DAMO method, but not quantification of urea by DAMO and NMR. All references to the SI figures need to be carefully checked, or relevant sentences reworded.
- (8) Lines 185-185: "At the relative positive potential region" -> "At more positive potentials within the range of ..."
- (9) Finally, the manuscript would strongly benefit from a careful proofread and improvements in the English grammar/style in quite a few places.

Reviewer 3.

Reviewer's Comments:

Zhao et al. have invested appropriate efforts into addressing the reviewers' comments from the first round, which produced a substantially improved manuscript, which can be recommended for publication subject to correction of the remaining minor, generally technical, issues. No further review is necessary from my side.

Response:

We sincerely thank you for your time and effort for our work. Your constructive suggestions are very helpful for us to improve the quality of this work, and we have revised our manuscript accordingly.

Question 1:

"Faraday efficiency" -> "faradaic efficiency" (see Bard's electrochemistry dictionary, if unsure).

Response:

Thank you very much for your suggestion, which is an important correction. We have also reconfirmed the formation of writing according to Bard's electrochemistry dictionary, and confirmed the version of "Faradaic efficiency". The corresponding parts of the manuscript have been revised.

Question 2:

The performance metrics should be reported in a statistically meaningful manner throughout the manuscript, and especially in the abstract and conclusions, i.e. as mean +/- standard deviation with appropriate number of significant figures.

Response:

The positive and negative deviation values have been added to the revised manuscript, and thank the reviewer for the comment.

Question 3:

Lines 47-48: "Moreover, the lower dissociation energy of the nitrogen-oxygen bond (204 kJ mol⁻¹) eases the coupling of NO₃⁻ reduction with CO₂ reduction to accomplish urea electrosynthesis from a thermodynamic perspective" – I would recommend removing the "thermodynamic perspective", as this is more about activation energy, i.e. kinetics.

Response:

Agreed with the reviewer's opinion, "from a thermodynamic perspective" was deleted in the revised manuscript.

Question 4:

Line 76: “reduction at a relative positive potential” – change to e.g. “positive potentials”, or better “practical overpotentials” or “low overpotentials”.

Response:

Thanks to the reviewer's comments, we have changed “a relative positive potential” to “low overpotentials”.

Question 5:

Line 133: “CuWO₄ and CuO increased significantly at the potential range from -0.1 to -0.4 V vs. RHE” - it is a bit of a stretch to refer to the observed increase as significant.

Better delete this word

Response:

The reviewer's comment was adopted, and we have deleted it according to the suggestion.

Question 6:

Figure captions should be much more detailed, and at minimum report the composition of the electrolyte solution and the gas used. For example, it is impossible to understand which experiments are shown in Figure S11, and quite a few other figures as well.

Response:

It is an oversight of an important detail. Thanks to the reviewer's comments, we have added more detailed experimental information to the Figure captions.

Question 7:

Lines 138-140: “and the production of urea was quantified by diacetylmonoxime-thiosemicarbazide (DAMO-TSC) and nuclear magnetic resonance (NMR) methods (Supplementary Fig. 12).” – Figure S12 shows a calibration for the DAMO method, but not quantification of urea by DAMO and NMR. All references to the SI figures need to be carefully checked, or relevant sentences reworded.

Response:

Thank you very much for the reviewer's careful reading. We have revised this part and checked the corresponding problems between the figures and texts in the manuscript.

Question 8:

Lines 185-185: “At the relative positive potential region” -> “At more positive potentials within the range of ...”

Response:

Thanks to the reviewer's comments. We have changed “At the relative positive potential region” to “At the low overpotential region” which might be more appropriate.

Question 9:

Finally, the manuscript would strongly benefit from a careful proofread and improvements in the English grammar/style in quite a few places.

Response:

We sincerely thank the reviewers for their suggestions, as well as the time and efforts they paid. We have checked and revised the whole manuscript in detail to show it with a more excellent state.